# $\mu$PC: Scaling Predictive Coding to 100+ Layer Networks

**Francesco Innocenti**
School of Engineering and Informatics
University of Sussex, UK
F.Innocenti@sussex.ac.uk

**El Mehdi Achour**
UM6P College of Computing
Rabat, Morocco
elmehdi.achour@um6p.ma

**Christopher L. Buckley**
School of Engineering and Informatics
University of Sussex, UK
VERSES AI Research Lab
Los Angeles, CA, USA
c.l.buckley@sussex.ac.uk

## Abstract

The biological implausibility of backpropagation (BP) has motivated many alternative, brain-inspired algorithms that attempt to rely only on local information, such as predictive coding (PC) and equilibrium propagation. However, these algorithms have notoriously struggled to train very deep networks, preventing them from competing with BP in large-scale settings. Indeed, scaling PC networks (PCNs) has recently been posed as a challenge for the community [48]. Here, we show that 100+ layer PCNs can be trained reliably using a Depth-$\mu$P parameterisation [72, 3] which we call "$\mu$PC". By analysing the scaling behaviour of PCNs, we reveal several pathologies that make standard PCNs difficult to train at large depths. We then show that, despite addressing only some of these instabilities, $\mu$PC allows stable training of very deep (up to 128-layer) residual networks on simple classification tasks with competitive performance and little tuning compared to current benchmarks. Moreover, $\mu$PC enables zero-shot transfer of both weight and activity learning rates across widths and depths. Our results serve as a first step towards scaling PC to more complex architectures and have implications for other local algorithms. Code for $\mu$PC is made available as part of a JAX library for PCNs.[1]

## 1 Introduction

Backpropagation (BP) is arguably the core algorithm behind the success of modern AI and deep learning [52, 29]. Yet, it is widely believed that the brain cannot implement BP due to its *non-local* nature [34], in that the update of any weight requires knowledge of all the weights deeper or further downstream in the network. This fundamental biological implausibility of BP has motivated the study of many local algorithms, including predictive coding (PC) [37, 36, 54, 63], equilibrium propagation [59, 74], and forward learning [20], among others [33, 43, 8]. These algorithms offer the potential for more energy efficient AI and have been argued to outperform BP in more biologically relevant

---

[1]https://github.com/thebuckleylab/jpc [23].

39th Conference on Neural Information Processing Systems (NeurIPS 2025).

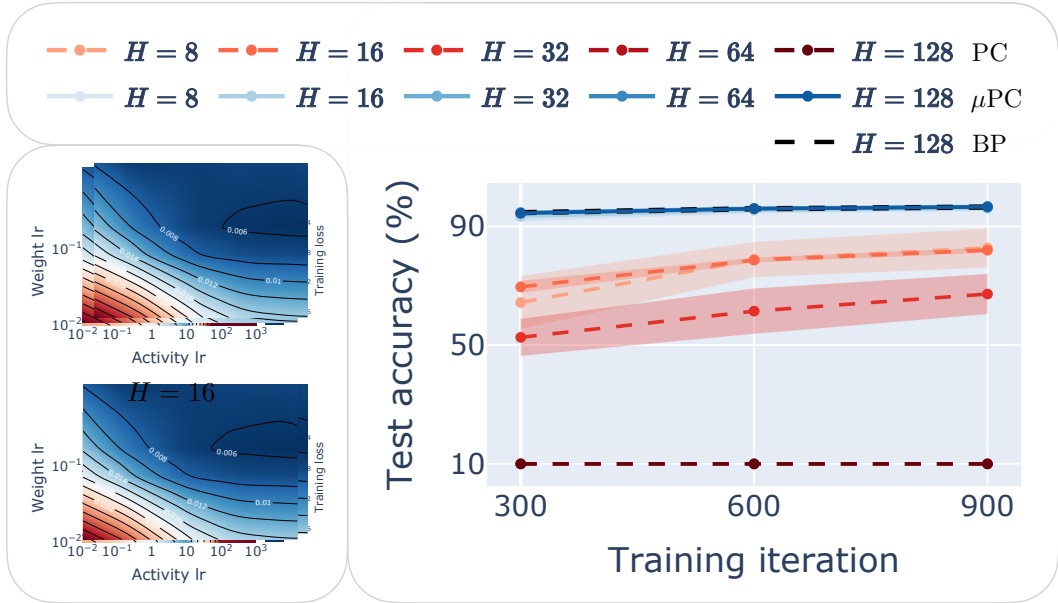

Figure 1: **μPC enables stable training of 100+ layer ResNets with zero-shot learning rate transfer.** (*Right*) Test accuracy of ReLU ResNets with depths $H = \{8, 16, 32, 64, 128\}$ trained to classify MNIST for one epoch with standard PC, μPC and BP with Depth-μP (see §A.4 for details). Solid lines and shaded regions indicate the mean and $\pm 1$ standard deviation across 3 different random seeds. These results hold across other activation functions (see Fig. A.16). See also Figs. A.17-A.19 for asymptotic results with 128-layer ReLU networks trained for multiple epochs on MNIST, Fashion-MNIST and CIFAR10. (*Left*) Example of zero-shot transfer of the weight and activity learning rates from 16- to 128-layer Tanh networks. See Figs. 5 & A.31-A.32 for an explanation and the complete transfer results across widths as well as depths.

settings such as online and continual learning [61]. However, local learning rules have notoriously struggled to train large and especially deep models on the scale of modern AI applications.[2]

For the first time, we show that very deep (100+ layer) networks can be trained reliably using a Depth-μP-inspired parameterisation [72, 3] of PC which we call "μPC" (Fig. 1). To our knowledge, *no networks of such depth have been trained before with a local algorithm.* Indeed, this has recently been posed as a challenge for the PC community [48]. We start by showing that the standard parameterisation of PC networks (PCNs) is inherently unscalable in that (i) the inference landscape becomes increasingly ill-conditioned with model size and training time, and (ii) the forward initialisation of the activities vanishes or explodes with the depth. We then show that, despite addressing only the second instability, μPC is capable of training up to 128-layer fully connected residual networks (ResNets) on standard classification tasks with competitive performance and little tuning compared to current benchmarks (Fig. 1). Moreover, μPC enables zero-shot transfer of both the weight and activity learning rates across widths and depths (Fig. 5). We make code for μPC available as part of a JAX library for PCNs at `https://github.com/thebuckleylab/jpc` [23].

The rest of the paper is structured as follows. Following a brief review of the maximal update parameterisation (μP) and PCNs (§2), Section 3 exposes two distinct pathologies in standard PCNs which make training at large scale practically impossible. Motivated by these findings, we then suggest a minimal set of desiderata for a more scalable PCN parameterisation (§4). Section 5 presents experiments with μPC, and Section 6 studies a specific regime where μPC converges to BP. We conclude with the limitations of this work and promising directions for future research (§7). For space reasons, we include related work and additional experiments in Appendix A, along with derivations, experimental details and supplementary figures.

---

[2]It is possible that these algorithms are more suited to alternative, non-digital hardware, but their scalability can still be investigated on standard GPUs. Indeed, the issues we expose with the standard parameterisation of PCNs can be argued to be hardware-independent (§3.1).

## 1.1 Summary of contributions

- We show that $\mu$PC, which reparameterises PCNs using Depth-$\mu$P [72, 3], allows stable training of very deep (100+ layer) ResNets on simple classification tasks with competitive performance and little tuning compared to current benchmarks [48] (Figs. 1 & A.17-A.18).

- $\mu$PC also empirically enables zero-shot transfer of both the weight and activity learning rates across widths and depths (Figs. 5 & A.31-A.32).

- We achieve these results by a theoretical and empirical analysis of the scaling behaviour of the inference landscape and dynamics of PCNs (§3), revealing the following two pathologies:
  - the inference landscape becomes increasingly ill-conditioned with model size (Fig. 2) and training time (Fig. 3) (§3.1); and
  - the forward pass of standard PCNs vanishes or explodes with the depth (§3.2).

- To address these instabilities, we propose a minimal set of desiderata that PCNs should aim to satisfy to be trainable at scale (§4), revealing an apparent trade-off between the conditioning of the inference landscape and the stability of the forward pass (Fig. 4). This analysis can be applied to other inference-based algorithms (§A.2.5).

- To better understand $\mu$PC, we study a theoretical regime where the $\mu$PC energy converges to the mean squared error (MSE) loss and so PC effectively implements BP (Theorem 1, Fig. 6). However, we find that $\mu$PC can successfully train deep networks far from this regime.

## 2 Background

### 2.1 The maximal update parameterisation ($\mu$P)

The maximal update parameterisation was first introduced by [70] to ensure that the order of the activation or feature updates at each layer remains stable with the width $N$. This was motivated by the lack of feature learning in the neural tangent kernel or "lazy" regime [27], where the activations remain practically unchanged during training [6, 31]. More formally, $\mu$P can be derived from the following 3 desiderata [70]: (i) the layer preactivations are $\mathcal{O}_N(1)$ at initialisation, (ii) the network output is $\mathcal{O}_N(1)$ during training, and (iii) the layer features are also $\mathcal{O}_N(1)$ during training.[3]

Satisfying these desiderata boils down to solving a system of equations for a set of scalars (commonly referred to as "abcd") parameterising the layer transformation, the (Gaussian) initialisation variance, and the learning rate [71, 44]. Different optimisers and types of layer lead to different scalings. One version of $\mu$P (and the version we will be using here) initialises all the weights from a standard Gaussian and rescales each layer transformation by $1/\sqrt{N_{\ell-1}}$, with the exception of the output which is scaled by $1/N_{L-1}$. Remarkably, $\mu$P allows not only for more stable training dynamics but also for *zero-shot hyperparameter transfer*: tuning a small model parameterised with $\mu$P guarantees that optimal hyperparameters such as the learning rate will transfer to a wider model [69, 42].

More recently, $\mu$P has been extended to depth for ResNets ("Depth-$\mu$P") [72, 3], such that transfer is also conserved across depths $L$. This is done by mainly introducing a $1/\sqrt{L}$ scaling before each residual block. Extensions of standard $\mu$P for other algorithms have also been proposed [25, 26, 14, 9].

### 2.2 Predictive coding networks (PCNs)

We consider the following general parameterisation of the energy function of $L$-layered PCNs [5]:

$$\mathcal{F} = \sum_{\ell=1}^{L} \frac{1}{2} ||\mathbf{z}_\ell - a_\ell \mathbf{W}_\ell \phi_\ell(\mathbf{z}_{\ell-1}) - \tau_\ell \mathbf{z}_{\ell-1}||^2 \tag{1}$$

with weights $\mathbf{W}_\ell \in \mathbb{R}^{N_\ell \times N_{\ell-1}}$, activities $\mathbf{z}_\ell \in \mathbb{R}^{N_\ell}$ and activation function $\phi_\ell(\cdot)$. Dense weight matrices could be replaced by convolutions, all assumed to be initialised i.i.d. from a Gaussian $(\mathbf{W}_\ell)_{ij} \sim \mathcal{N}(0, b_\ell)$ with variance scaled by $b_\ell$. We omit multiple data samples to simplify the notation, and ignore biases since they do not affect the main analysis, as explained in §A.2.1. We also add scalings $a_\ell \in \mathbb{R}$ and optional skip or residual connections set by $\tau_\ell \in \{0, 1\}$.

---

[3]Throughout, we will use $\mathcal{O}_n(1)$ to mean $\Theta_n(1)$ such that the activations neither explode nor vanish with $n$.

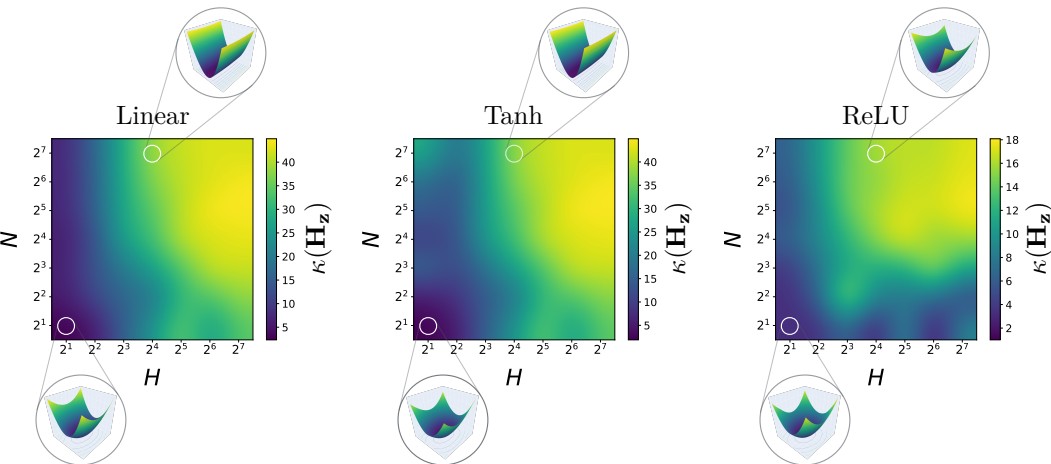

Figure 2: **Wider and particulary deeper PCNs have a more ill-conditioned inference landscape.** We plot the condition number of the activity Hessian $\kappa(\mathbf{H_z})$ (lower is better) of randomly initialised fully connected networks as a function of the width $N$ and depth $H$ (see §A.4 for details). Insets show 2D projections of the landscape of selected networks around the linear solution (Eq. 4) along the maximum and minimum eigenvectors of the Hessian $\mathcal{F}(\mathbf{z}^* + \alpha\hat{\mathbf{v}}_{\min} + \beta\hat{\mathbf{v}}_{\max})$. Note that the ill-conditioning is much more extreme for ResNets (see Fig. A.22). Results were similar across different seeds.

The energy of the last layer is defined as $\mathcal{F}_L = \frac{1}{2}||\mathbf{z}_L - a_L\mathbf{W}_L\phi_L(\mathbf{z}_{L-1})||^2$ for some target $\mathbf{z}_L := \mathbf{y} \in \mathbb{R}^{d_{\text{out}}}$, while the energy of the first layer is $\mathcal{F}_1 = \frac{1}{2}||\mathbf{z}_1 - a_1\mathbf{W}_1\mathbf{z}_0||^2$, with some optional input $\mathbf{z}_0 := \mathbf{x} \in \mathbb{R}^{d_{\text{in}}}$ for supervised (vs unsupervised) training.[4] We will refer to PC or SP as the "standard parameterisation" with unit premultipliers $a_\ell = 1$ for all $\ell$ and standard initialisations [30, 11, 18] such as $b_\ell = 1/N_{\ell-1}$, and to $\mu$PC as that which uses (some of) the scalings of Depth-$\mu$P (§2.1).[5] See Table 1 for a summary.

We fix the width of all the hidden layers $N = N_1 = \cdots = N_H$ where $H = L - 1$ is the number of hidden layers. We use $\boldsymbol{\theta} := \{\text{vec}(\mathbf{W}_\ell)\}_{\ell=1}^L \in \mathbb{R}^p$ to represent all the weights with $p$ as the total number of parameters and $\mathbf{z} := \{\mathbf{z}_\ell\}_{\ell=1}^H \in \mathbb{R}^{NH}$ to denote all the activities free to vary. Note that, depending on the context, we will use both $H$ and $L$ to refer to the network depth.

PCNs are trained by minimising the energy (Eq. 1) in two separate phases: first with respect to the activities (inference) and then with respect to the weights (learning),

$$\textit{Infer:} \quad \min_{\mathbf{z}} \mathcal{F} \qquad (2) \qquad\qquad \textit{Learn:} \quad \min_{\boldsymbol{\theta}} \mathcal{F}. \qquad (3)$$

Inference acts on a single data point and is generally performed by gradient descent (GD), $\mathbf{z}_{t+1} = \mathbf{z}_t - \beta\nabla_{\mathbf{z}}\mathcal{F}$ with step size $\beta$. The weights are often updated at numerical convergence of the inference dynamics, when $\nabla_{\mathbf{z}}\mathcal{F} \approx 0$. Our theoretical results will mainly address the first optimisation problem (Eq. 2), namely the inference landscape and dynamics, but we discuss and numerically investigate the impact on the learning dynamics (Eq. 3) wherever relevant.

## 3 Instability of the standard PCN parameterisation

In this section, we reveal through both theory and experiment that the standard parameterisation (SP) of PCNs suffers from two instabilities that make training and convergence of the PC inference dynamics (Eq. 2) at large scale practically impossible. First, the inference landscape of standard PCNs becomes increasingly ill-conditioned with model size and training time (§3.1). Second, depending

---

[4]Many of our theoretical results can be extended to the unsupervised case (see §A), but for ease of presentation we will focus on the supervised case.

[5]We distinguish between $\mu$PC and Depth-$\mu$P for brevity, to encapsulate both the algorithm and the parameterisation in a single acronym.

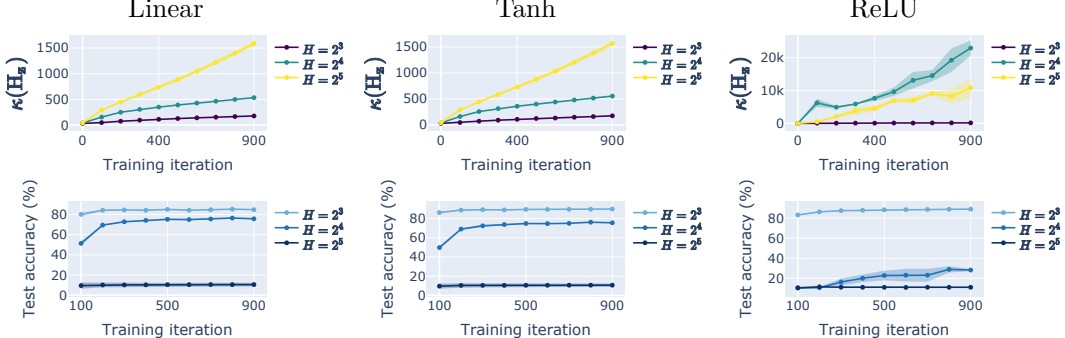

Figure 3: **The inference landscape of PCNs grows increasingly ill-conditioned with training.** We plot the condition number of the activity Hessian (Eq. 5) (*top*) as well as test accuracies (*bottom*) for fully connected networks of depths $H \in \{8, 16, 32\}$ during one epoch of training. All networks had width $N = 128$ and were trained to classify MNIST (see §A.4 for more details). Similar results are observed for ResNets (Fig. A.9) and Fashion-MNIST (Fig. A.23). Solid lines and shaded regions indicate the mean and standard deviation over 3 random seeds.

on the model, the feedforward pass either vanishes or explodes with the depth (§3.2). The second problem is shared with BP-trained networks, while the first instability is unique to PC and likely any other algorithm performing inference minimisation (§A.2.5).

## 3.1 Ill-conditioning of the inference landscape

Here we show that the inference landscape of standard PCNs becomes increasingly ill-conditioned with network width, depth and training time. As reviewed in §2.2, the inference phase of PC (Eq. 2) is commonly performed by GD. For a deep linear network (DLN, Eq. 1 with $\phi_\ell = \mathbf{I}$ for all $\ell$), one can solve for the activities in closed form as shown by [26],

$$\nabla_{\mathbf{z}} \mathcal{F} = \mathbf{H}_{\mathbf{z}} \mathbf{z} - \mathbf{b} = 0 \quad \Longrightarrow \quad \mathbf{z}^* = \mathbf{H}_{\mathbf{z}}^{-1} \mathbf{b} \tag{4}$$

where $(\mathbf{H}_{\mathbf{z}})_{\ell k} \coloneqq \partial^2 \mathcal{F}/\partial \mathbf{z}_\ell \partial \mathbf{z}_k \in \mathbb{R}^{(NH) \times (NH)}$ is the Hessian of the energy with respect to the activities, and $\mathbf{b} \in \mathbb{R}^{NH}$ is a sparse vector depending only on the data and associated weights (see §A.2.1 for details). Eq. 4 shows that for a DLN, PC inference is a well-determined linear problem.[6]

For arbitrary DLNs, one can also prove that the inference landscape is strictly convex as the Hessian is positive definite[7], $\mathbf{H}_{\mathbf{z}} \succ 0$ (Theorem A.1; see §A.2.2 for proof). This makes intuitive sense since the energy (Eq. 1) is quadratic in $\mathbf{z}$. The result is empirically verified for DLNs in Figs. A.5-A.7 and appears to generally hold for nonlinear networks (see Figs. A.7 & A.22).

For such convex problems, the convergence rate of GD is known to be given by the condition number of the Hessian [4, 41], $\kappa(\mathbf{H}_{\mathbf{z}}) = |\lambda_{\max}|/|\lambda_{\min}|$. Intuitively, the higher the condition number, the more elliptic the level sets of the energy $\mathcal{F}(\mathbf{z})$ become, and the more iterations GD will need to reach the solution (see Fig. A.21), with the step size bounded by the highest curvature direction $\beta < 2/\lambda_{\max}$ (see Fig. A.10 for an example). For non-convex problems, it can still be useful to have a notion of local conditioning [e.g. 73].

What determines the condition number of $\mathbf{H}_{\mathbf{z}}$? Looking more closely at the structure of the Hessian

$$\frac{\partial^2 \mathcal{F}}{\partial \mathbf{z}_\ell \partial \mathbf{z}_k} = \begin{cases} \mathbf{I} + a_{\ell+1}^2 \mathbf{W}_{\ell+1}^T \mathbf{W}_{\ell+1}, & \ell = k \\ -a_{k+1} \mathbf{W}_{k+1}, & \ell - k = 1 \\ -a_{\ell+1} \mathbf{W}_{\ell+1}^T, & \ell - k = -1 \\ \mathbf{0}, & \text{else} \end{cases}, \tag{5}$$

---

[6]This contrasts with the weight landscape $\mathcal{F}(\boldsymbol{\theta})$, which grows nonlinear with the depth even for DLNs [22].

[7]We note that this was claimed to be proved by [39]; however, they only showed that the block diagonals of the Hessian are positive definite, ignoring the layer, off-diagonal interactions.

one realises that it depends on two main factors: (i) the *network architecture*, including the width $N$, depth $L$ and connectivity; and (ii) the *value of the weights* at any time during training $\boldsymbol{\theta}_t$. We first find that the inference landscape of standard PCNs becomes increasingly ill-conditioned with the width and particularly depth (Fig. 2), and extremely so for ResNets (Fig. A.22). See also §A.2.3 for a random matrix theory analysis of the scaling behaviour of the initialised Hessian eigenspectrum with $N$ and $L$. In addition, we observe that the ill-conditioning grows and spikes during training (Figs. 3, A.9, A.23 & A.25), and using an adaptive optimiser such as Adam [28] does not seem to help (Figs. A.8 & A.24). Together, these findings help to explain why the convergence of the GD inference dynamics (Eq. 2) can dramatically slow down on deeper models [23, 48], while also highlighting that small inference gradients—which are commonly used to determine convergence—do not necessarily imply closeness to a solution.

### 3.2 Vanishing/exploding forward pass

In the previous section (§3.1), we saw that the growing ill-conditioning of the inference landscape with the model size and training time is one likely reason for the challenging training of PCNs at large scale. Another reason—and as we will see the key reason—is that the forward initialisation of the activities can vanish or explode with the depth. This is a classic finding in the neural network literature that has been surprisingly ignored for PCNs. For fully connected networks with standard initialisations [30, 11, 18], the forward pass vanishes with the depth, leading to vanishing gradients. This issue can be addressed with residual connections [19] and various forms of activity normalisation [24, 1], both of which remain key components of the modern transformer block [64].

However, while there have been attempts to train ResNets with PC [48], they have been without activity normalisation. This is likely because any kind of normalisation of the activities seems at odds with convergence of the inference dynamics to a solution (Eq. 2). Without normalisation, however, the activations (and gradients) of vanilla ResNets explode with the depth (see Fig. A.30). A potential remedy would be to normalise only the forward pass, but here we will aim to take advantage of more principled approaches with stronger guarantees about the stability of the forward pass (§4).

## 4 Desiderata for stable PCN parameterisation

In §3, we exposed two main pathologies in the scaling behaviour of standard PCNs: (i) the growing ill-conditioning of the inference landscape with model size and training time (§3.1), and (ii) the instability of the forward pass with depth (§3.2). These instabilities motivate us to specify a minimal set of *desiderata* that we would like a PCN to satisfy to be trainable at large scale.[8]

> **Desideratum 1.** *Stable forward pass at initialisation.* At initialisation, all the layer preactivations are stable independent of the network width and depth, $||\mathbf{z}_\ell|| \sim \mathcal{O}_{N,H}(1)$ for all $\ell$, where $\mathbf{z}_\ell = h_\ell(\dots h_1(\mathbf{x}))$ with $h_\ell(\cdot)$ as the map relating one layer to the next.

To our knowledge, there are two approaches that provide strong theoretical guarantees about this desideratum: (i) orthogonal weight initialisation for both fully connected [58, 46, 47, 68] and convolutional networks [68], ensuring that $\mathbf{W}_\ell^T \mathbf{W}_\ell = \mathbf{I}$ at every layer $\ell$; and (ii) the recent Depth-$\mu$P parameterisation [72, 3] (see §2.1 for a review). For a replication of these results, see Fig. A.30. To apply Depth-$\mu$P to PC, we simply reparameterise the PC energy for ResNets (Eq. 1 with $\tau_\ell = 1$ for $\ell = 2, \dots, H$ and $\tau_\ell = 0$ otherwise) with the layer scalings of Depth-$\mu$P (see Table 1).[9] We call this reparameterisation $\mu$PC.

---

[8]We do not see these desiderata as strict (necessary or sufficient) conditions, since relatively small PCNs can be trained competitively without satisfying them, and other conditions might be needed for successful training.

[9]$\mu$P and Depth-$\mu$P also include an optimiser-dependent scaling of the learning rate. However, we found this scaling to be suboptimal for PC as discussed in §7.

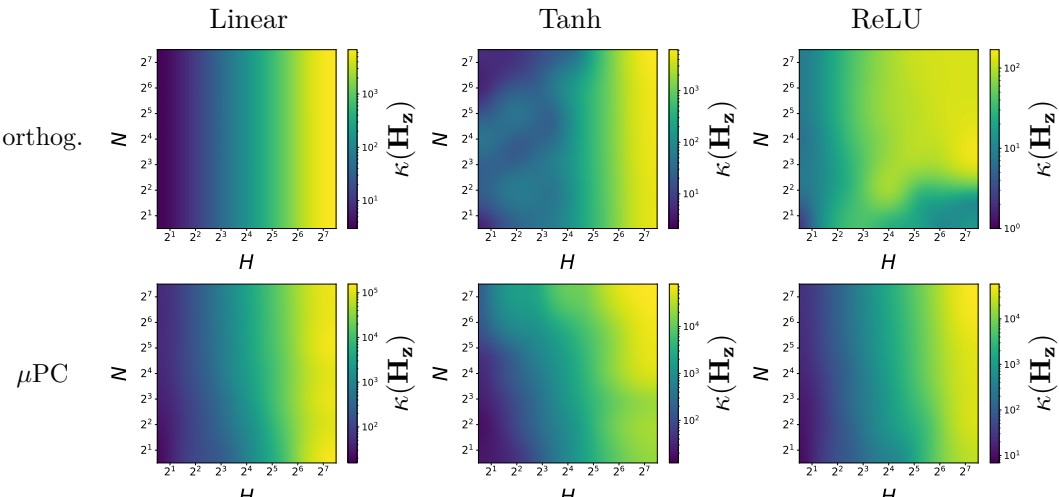

Figure 4: **Parameterisations with stable forward passes induce highly ill-conditioned inference landscapes with depth.** We plot the conditioning of the activity Hessian of randomly initialised networks over width $N$ and depth $H$ for the $\mu$PC and orthogonal parameterisations. Networks with and without residual connections were used for these respective parameterisations. Note that ReLU networks with orthogonal initialisation cannot achieve stable forward passes (see Fig. A.30). Results were similar across different seeds.

Table 1: **Summary of parameterisations.** Standard PC has unit layer premultipliers and weights initialised from a Gaussian with variance scaled by the input width at every layer $N_{\ell-1}$. $\mu$PC uses a standard Gaussian initialisation and adds width- and depth-dependent scalings at every layer.

|        | $a_1$ (input weights) | $a_\ell$ (hidden weights) | $a_L$ (output weights) | $b_\ell$ (init. variance) |
|--------|-----------------------|---------------------------|------------------------|---------------------------|
| PC     | 1                     | 1                         | 1                      | $N_{\ell-1}^{-1}$         |
| $\mu$PC | $N_0^{-1/2}$         | $(N_{\ell-1}L)^{-1/2}$    | $N_{L-1}^{-1}$         | 1                         |

We would like Desideratum 1 to hold throughout training as we state in the following desideratum.

> **Desideratum 2.** *Stable forward pass during training.* The forward pass is stable during training such that Desideratum 1 is true for all training steps $t = 1, \ldots, T$.

Depth-$\mu$P ensures this desideratum for BP, but we do not know whether the same will apply to $\mu$PC. We return to this point in §7. For the orthogonal parameterisation, the weights should remain orthogonal during training to satisfy Desideratum 2, which could be encouraged with some kind of regulariser. Next, we address the ill-conditioning of the inference landscape (§3.1), again first at initialisation.

> **Desideratum 3.** *Stable conditioning of the inference landscape at initialisation.* The condition number of the activity Hessian (Eq. 5) at initialisation stays constant with the network width and depth, $\kappa(\mathbf{H_z}) \sim \mathcal{O}_{N,H}(1)$.

Ideally, we would like the PC inference landscape to be perfectly conditioned, i.e. $\kappa(\mathbf{H_z}) = 1$. However, this cannot be achieved without zeroing out the weights, $\mathbf{H_z}(\boldsymbol{\theta} = \mathbf{0}) = \mathbf{I}$, since the Hessian is symmetric and so it can only have all unit eigenvalues if it is the identity. Starting with small weights $(\mathbf{W}_\ell)_{ij} \ll 1$ at the cost of slightly imperfect conditioning is not a solution, since the forward pass vanishes, thus violating Desideratum 1. See §A.3.3 for another intervention that appears to come at the expense of performance.

What about the above parameterisations ensuring stable forward passes? Interestingly, both orthogonal initialisation and $\mu$PC induce highly ill-conditioned inference landscapes with the depth (Fig. 4), similar to standard PC ResNets (Fig. A.22). This highlights a potential trade-off between the stability of the forward pass (technically, the conditioning of the input-output Jacobian) and the conditioning of the activity Hessian. Because PCNs with ill-conditioned inference landscapes can still be trained (e.g. see Fig. 3), we will choose to satisfy Desideratum 1 at the expense of Desideratum 3, while seeking to prevent the condition number from exploding during training.

> **Desideratum 4.** *Stable conditioning of the inference landscape during training.* The condition number of the activity Hessian (Eq. 5) is stable throughout training such that $\kappa(\mathbf{H_z}(t)) \approx \kappa(\mathbf{H_z}(t-1))$ for all training steps $t = 1, \ldots, T$.

## 5 Experiments

We performed experiments with parameterisations ensuring stable forward passes at initialisation (Desideratum 1), namely $\mu$PC and orthogonal, despite their inability to solve the ill-conditioning of the inference landscape with depth (Desideratum 3; Fig. 4). Due to limited space, we report results only for $\mu$PC since orthogonal initialisation was not found to be as effective (see §A.3.4). We trained fully connected residual PCNs on standard image classification tasks (MNIST, Fashion-MNIST and CIFAR10). This simple setup was chosen because the main goal was to test whether $\mu$PC is capable of training deep PCNs—a task that has proved challenging with more complex datasets and architectures [48]. We note that all the networks used as many inference steps as hidden layers (see Figs. A.14 & A.27 for results with one step).

First, we trained ResNets of varying depth (up to 128 layers) to classify MNIST for a single epoch. Remarkably, we find that $\mu$PC allows stable training of networks of all depths across different activation functions (Figs. 1 & A.16). These networks were tuned only for the weight and activity learning rates, with no other optimisation techniques such as momentum, weight decay, and nudging, as used in previous studies [48]. Competitive performance ($\approx 98\%$) is achieved in 5 epochs (Fig. A.17), $5\times$ faster than the current benchmark [48]. Similar results are observed on Fashion-MNIST, where competitive accuracy ($\approx 89\%$) is reached in fewer than 15 epochs (Fig. A.18). On CIFAR10, performance is far from SOTA because of the fully connected (as opposed to convolutional) architectures used, but $\mu$PC remains trainable at large depth (Fig. A.19).

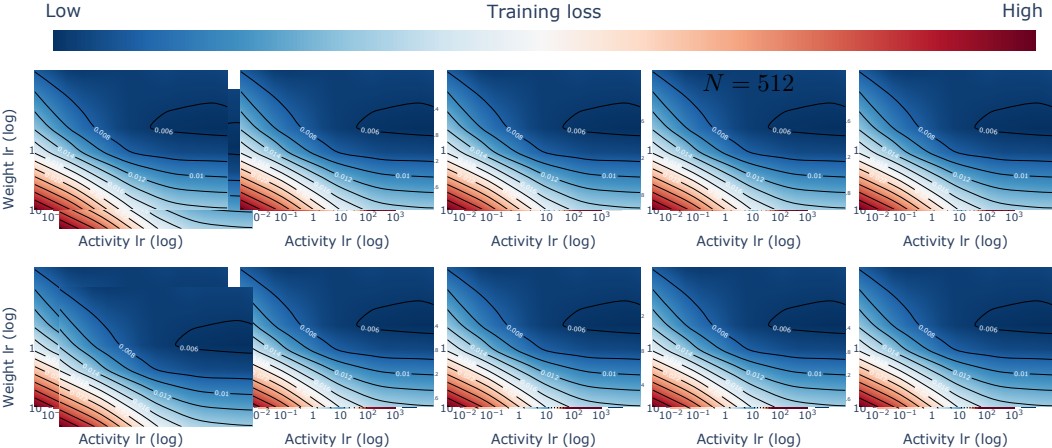

Figure 5: $\mu$**PC enables zero-shot transfer of the weight and activity learning rates across widths** $N$ **and depths** $H$**.** Minimum training loss (log) achieved by ResNets of varying width and depth trained with $\mu$PC on MNIST across different weight and activity learning rates. All networks had Tanh as nonlinearity (see Figs. A.31-A.32 for other activation functions); those with varying width (first row) had 8 hidden layers, and those with varying the depth (second row) had 512 hidden units (see §A.4 for details). Each contour was averaged over 3 random seeds.

Strikingly, we also find that $\mu$PC enables zero-shot transfer of both the weight and activity learning rates across widths and depths (Figs. 5 & A.31-A.32), consistent with recent results with Depth-$\mu$P [72, 3]. This means that one can tune a small PCN and then transfer the optimal learning rates to wider and/or deeper PCNs—a process that is particularly costly for PC since it requires two separate learning rates. In fact, this is precisely how we obtained the Fashion-MNIST (Fig. A.18) and CIFAR10 (Fig. A.19) results: by performing transfer from 8- to 128-layer networks, avoiding the expensive tuning at large scale.

## 6 Is $\mu$PC BP?

Why does $\mu$PC seem to work so well despite failing to solve the ill-conditioning of the inference landscape with depth (Fig. 4)? Depth-$\mu$P also satisfies other, BP-specific desiderata that PC might not require or benefit from. Here we show that while there is a practical regime where $\mu$PC approximates BP, it turns out to be brittle, and so BP cannot explain the success of $\mu$PC (at least on the tasks considered). In particular, it is possible to show that, when the width is much larger than the depth $N \gg L$, at initialisation the $\mu$PC energy at the inference equilibrium converges to the MSE loss. In this regime, PC computes the same gradients as BP and all the Depth-$\mu$P theory applies.

> **Theorem 1** (Limit Convergence of $\mu$PC to BP.). *Let $\mathcal{F}_{\mu PC}(\boldsymbol{\theta}, \mathbf{z})$ be the PC energy of a randomly initialised linear ResNet (Eq. 1 with $\tau_\ell = 1$ for $\ell = 2, \ldots, H$ and $\tau_\ell = 0$ otherwise) parameterised with Depth-$\mu$P (Table 1) and $\mathcal{L}_{\mu P}(\boldsymbol{\theta})$ its corresponding MSE loss. Then, as the aspect ratio of the network $r := L/N$ vanishes, the equilibrated energy (Eq. 31) converges to the loss (see §A.2.6 for proof)*
> $$r \to 0, \quad \mathcal{F}_{\mu PC}(\boldsymbol{\theta}, \mathbf{z}^*) = \mathcal{L}_{\mu P}(\boldsymbol{\theta}). \tag{6}$$

The result relies on a recent derivation of the equilibrated energy as a rescaled MSE loss for DLNs [22]. We simply extend this to linear ResNets and show that the rescaling approaches the identity with $\mu$PC in the above limit. Fig. 6 shows that the result holds at initialisation ($t = 0$), with the equilibrated energy converging to the loss when the width is around $32\times$ the depth. (Note that the deepest networks ($H = 128, N = 512$) we tested in the previous section had a much smaller aspect ratio, $r = 4$.) Nevertheless, we observe that the equilibrated energy starts to diverge from the loss with training at large width and depth (Fig. 6). Note also that we do not know the inference solution for nonlinear networks. We therefore leave further theoretical study of $\mu$PC to future work. See also §A.1 for a discussion of how Theorem 1 relates to previous correspondences between PC and BP.

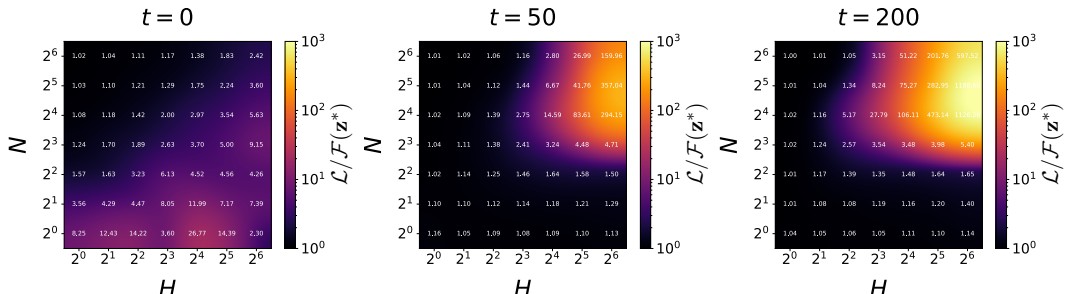

Figure 6: **Convergence/Divergence of $\mu$PC to BP for linear ResNets.** To verify Theorem 1 (Eq. 6), we plot the ratio between the MSE loss and the equilibrated $\mu$PC energy of linear ResNets (Eq. 31) at different training points $t$ as a function of the width $N$ and depth $H$ (see §A.4 for details). We observe that while at initialisation ($t = 0$) the equilibrated energy converges to the loss as the the width grows relative to the depth (verifying Theorem 1), the correspondence breaks down with training at large depth and width. Results were similar across different runs.

# 7 Discussion

In summary, we showed that it is possible to reliably train very deep (100+ layer) networks with a local learning algorithm. We achieved this via a Depth-$\mu$P-like reparameterisation of PCNs which we labelled $\mu$PC. We found that $\mu$PC is capable of training very deep networks with little tuning and competitive performance on simple classification tasks (Fig. 1), while also enabling zero-shot transfer of weight and activity learning rates across widths and depths (Fig. 5).

**$\mu$PC and inference ill-conditioning.** Despite its relative success, $\mu$PC failed to solve the growing ill-conditioning of the inference landscape with the network depth (Desideratum 3; Fig. 4). This can be explained by two additional findings. First, the forward pass of $\mu$PC seems to initialise the activities much closer to the analytical solution (Eq. 4) for DLNs than standard PC (Fig. A.35). Second, training $\mu$PC networks with a single inference step (as opposed to as many as hidden layers) led to performance degradation not only during training, but also with depth (Figs. A.14 & A.27). Together, these results suggest that a stable forward pass, as ensured by $\mu$PC, is critical not only for performance but also for dealing with landscape ill-conditioning, by initialising the activities closer to a solution such that only a few (empirically determined) inference steps are needed. This is also consistent with the finding that while inference convergence is necessary for successful training of the SP, it does not appear sufficient for good generalisation (see §A.3.6). It would be interesting to study $\mu$PC in more detail in linear networks given their analytical tractability.

Another recent study investigated the problem of training deep PCNs [12], showing an exponential decay in the activity gradients over depth. This result can be seen as a consequence of the ill-conditioning of the inference landscape with depth (Fig. 2), since flat regions where the forward pass seems to initialise the activities (see §A.3.2) have small gradients, and depth drives ill-conditioning. [12] proposed a reparameterisation of PCNs leveraging BP for faster inference convergence on GPUs, and it could be interesting to combine this approach with $\mu$PC, especially for generation tasks or more complex datasets where more inference steps might be necessary for good performance.

**$\mu$PC and the other Desiderata.** Did $\mu$PC satisfy some other Desiderata (§4) besides the stability of the forward pass at initialisation (Desideratum 1)? When experimenting with $\mu$PC, we tried including the Depth-$\mu$P scalings only in the forward pass (i.e. removing them from the energy or even just the inference or weight gradients). However, this always led to non-trainable networks even at small depths, suggesting that the Depth-$\mu$P scalings are also beneficial for the PC inference and learning dynamics and that the resulting updates are likely to keep the forward pass stable during training (Desideratum 2). Deriving principled scalings specific to PC could help explain these findings or even lead to better scalings. Finally, $\mu$PC did not seem to prevent the ill-conditioning of the inference landscape from growing with training (see Figs. A.28 & A.29), thus violating Desideratum 4.

**Is $\mu$PC optimal?** $\mu$PC unlikely to be the optimal parameterisation for PCNs. This is because we adapted, rather than derived, principled (Depth-$\mu$P) scalings for BP, with only guarantees about the stability of the forward pass. Indeed, we did not rescale the learning rate of Adam (used in all our experiments) by $\sqrt{NL}$ as prescribed by Depth-$\mu$P [72], since this scaling always led to non-trainable networks. We note that depth transfer has also been achieved without this scaling [3, 42] and that the optimal depth scaling is still an active area of research [10]. It would also be useful to better understand the relationship between $\mu$PC and the (width-only) $\mu$P parameterisation for PC proposed by [26] (see §A.1 for a comparison). More generally, it would therefore be potentially impactful to derive principled scalings specific to PC. While an analysis far from inference equilibrium appears challenging, one could start with the order of the weight updates of the equilibrated energy of linear ResNets (Eq. 31).

**Other future directions.** Given the recent successful application of Depth-$\mu$P to convolutional networks and transformers [3, 42], it would be interesting to investigate whether these more complex architectures can be successfully trained on large-scale datasets with $\mu$PC. Our analysis of the inference landscape can also be applied to any other algorithm performing some kind of inference minimisation (see §A.2.5 for a preliminary investigation of equilibrium propagation), and it could be interesting to see whether these algorithms could also benefit from $\mu$P-like parameterisation.

## Acknowledgements

FI is funded by the Sussex Neuroscience 4-year PhD Programme. EMA acknowledges funding by UM6P and the Deutsche Forschungsgemeinschaft (DFG, German Research Foundation) - Project number 442047500 through the Collaborative Research Center "Sparsity and Singular Structures" (SFB 1481) as he started this project at RWTH Aachen University. CLB was partially supported by the European Innovation Council (EIC) Pathfinder Challenges, Project METATOOL with Grant Agreement (ID: 101070940). FI would like to thank Alexandru Meterez and Lorenzo Noci for their help in better understanding $\mu$P, and Ivor Simpson for providing access to GPUs used to run some of the experiments.

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

# A  Appendix

**Contents**

## A.1  Related work

$\mu$**P for PC [26].**  The study closest to our work is [26], who derived a $\mu$P parameterisation for PC (as well as target propagation), also showing hyperparameter transfer across widths. This work differs from ours in the following three important aspects: (i) it derives $\mu$P for PC only for the width, (ii) it focuses on regimes where PC approximates or is equivalent to other algorithms (including BP) so that all the $\mu$P theory can be applied, and (iii) it considers layer-wise scalar precisions $\gamma_\ell$ for each layer energy term, which are not standard in how PCNs are trained (but are nevertheless interesting to study). By contrast, we propose to apply Depth-$\mu$P to PC, showing transfer for depth as well as width (Figs. 5 & A.31-A.32). We also study a regime where this parameterisation reduces to BP (Fig. 6) while showing that successful training is still possible far from this regime (Fig. 1).

**Training deep PCNs [49, 48].**  Our work is also related to [49], who following [48] showed that the PC energy (Eq. 1) is disproportionately concentrated at the output layer $\mathcal{F}_L$ (closest to the target) for deep PCNs. They conjecture that this is problematic for two reasons: first, it does not allow the model to use (i.e. update) all of its layers; and second, it makes the latents diverge from the forward pass, which they claim leads to suboptimal weight updates. The first point is consistent with our theory and experiments. In particular, because the activities of standard PCNs vanish or explode with the depth (§3.2) and stay almost constant during inference due to the ill-conditioning of the landscape (§3.1) (Figs. A.10-A.11 & A.36), the weight updates are likely to be imbalanced across layers. However, the ill-conditioning contradicts the second point, in that the activities barely move during inference and stay close to the forward pass (see §A.3.2 for relevant experiments). Moreover, divergence from the forward pass does not necessarily lead to suboptimal weight updates and worse performance. For standard PC, deep networks cannot achieve good performance regardless of whether one stays close to the forward pass (see §A.3.6). For $\mu$PC, on the other hand, as many steps as the number of hidden layers (e.g. Fig. 1) leads to depth-stable and much better accuracy than a single step (e.g. Fig. A.14).

**PC and BP.** Our theoretical result about the convergence of $\mu$PC to BP (Theorem 1) relates to a relatively well-established series of correspondences between PC and BP [66, 40, 60, 51, 56, 38]. In brief, if one makes some rather biologically implausible assumptions (such as precisely timed inference updates), it can be shown that PC can approximate or even compute exactly the same gradients as BP. In stark contrast to these results and also the work of [26] (which requires arbitrarily specific precision values at different layers), Theorem 1 applies to standard PC, with arguably interpretable width- and depth-dependent scalings.[10]

**Theory of PC inference (Eq. 2) & learning (Eq. 3).** Finally, our work can be seen as a companion paper to [22], who provided the first rigorous, explanatory and predictive theory of the learning landscape and dynamics of practical PCNs (Eq. 3). They first show that for DLNs the energy at the inference equilibrium is a rescaled MSE loss with a weight-dependent rescaling, a result that we build on here for Theorem 1. They then characterise the geometry of the equilibrated energy (the effective landscape on which PC learns), showing that many highly degenerate saddles of the loss including the origin become much easier to escape in the equilibrated energy. Here, by contrast, we focus on the geometry of the *inference landscape and dynamics* (Eq. 2). As an aside, we note that the origin saddle result of [22] probably breaks down for ResNets, where for the linear case it has been shown that the saddle is effectively shifted and the origin becomes locally convex [15]. We suspect that the results generalise, but it could still be interesting to extend the theory of [22] to ResNets, especially by also looking at the geometry of minima.

**$\mu$P.** For a full treatment of $\mu$P and its extensions, we refer the reader to key works of the "Tensor Programs" series [70, 69, 71, 72]. $\mu$P effectively puts feature learning back into the infinite-width limit of neural networks, lacking from the neural tangent kernel (NKT) or "lazy" regime [27, 6, 31]. In particular, in the NTK the layer preactivations evolve in $\mathcal{O}(N^{-1/2})$ time. In $\mu$P, the features instead change in a "maximal" sense (hence "$\mu$"), in that they vary as much as possible without diverging with the width, which occurs for the output predictions under SP [70]. More formally, $\mu$P can be derived from the 3 desiderata stated in §2.1. $\mu$P was extended to depth (Depth-$\mu$P) for ResNets by mainly introducing a $1/\sqrt{L}$ scaling before each residual block [72, 3]. This breakthrough was enabled by the commutativity of the infinite-width and infinite-depth limit of ResNets [17, 16]. Standard $\mu$P has also been extended to local algorithms including PC [26] (see $\mu$**P for PC** above), sparse networks [9], second-order methods [25], and sharpness-aware minimisation [14].

### A.2 Proofs and derivations

All the theoretical results below are derived for linear networks of some form.

### A.2.1 Activity gradient (Eq. 4) and Hessian (Eq. 5) of DLNs

The gradient of the energy with respect to all the PC activities of a DLN (Eq. 4) can be derived by simple rearrangement of the partials with respect to each layer, which are given by

$$\partial \mathcal{F}/\partial \mathbf{z}_1 = \mathbf{z}_1 - a_1 \mathbf{W}_1 \mathbf{x} - a_2 \mathbf{W}_2^T \mathbf{z}_2 + a_2^2 \mathbf{W}_2^T \mathbf{W}_2 \mathbf{z}_1 \tag{7}$$

$$\partial \mathcal{F}/\partial \mathbf{z}_2 = \mathbf{z}_2 - a_2 \mathbf{W}_2 \mathbf{z}_1 - a_3 \mathbf{W}_3^T \mathbf{z}_3 + a_3^2 \mathbf{W}_3^T \mathbf{W}_3 \mathbf{z}_2 \tag{8}$$

$$\vdots \tag{9}$$

$$\partial \mathcal{F}/\partial \mathbf{z}_H = \mathbf{z}_H - a_{L-1} \mathbf{W}_{L-1} \mathbf{z}_{H-1} - a_L \mathbf{W}_L^T \mathbf{y} + a_L^2 \mathbf{W}_L^T \mathbf{W}_L \mathbf{z}_H. \tag{10}$$

Factoring out the activity of each layer

$$\partial \mathcal{F}/\partial \mathbf{z}_1 = \mathbf{z}_1(\mathbf{1} + a_2^2 \mathbf{W}_2^T \mathbf{W}_2) - a_1 \mathbf{W}_1 \mathbf{x} - a_2 \mathbf{W}_2^T \mathbf{z}_2 \tag{11}$$

$$\partial \mathcal{F}/\partial \mathbf{z}_2 = \mathbf{z}_2(\mathbf{1} + a_3^2 \mathbf{W}_3^T \mathbf{W}_3) - a_2 \mathbf{W}_2 \mathbf{z}_1 - a_3 \mathbf{W}_3^T \mathbf{z}_3 \tag{12}$$

$$\vdots \tag{13}$$

$$\partial \mathcal{F}/\partial \mathbf{z}_H = \mathbf{z}_H(\mathbf{1} + a_L^2 \mathbf{W}_L^T \mathbf{W}_L) - a_{L-1} \mathbf{W}_{L-1} \mathbf{z}_{H-1} - a_L \mathbf{W}_L^T \mathbf{y}, \tag{14}$$

---

[10]The width scaling is inherently local, while the depth scaling is more global but could be perhaps argued to be bio-plausible based on a notion of the brain "knowing its own depth".

one realises that this can be rearranged in the form of a linear system

$$
\nabla_{\mathbf{z}}\mathcal{F} = \underbrace{\begin{bmatrix}
\mathbf{I} + a_2^2\mathbf{W}_2^T\mathbf{W}_2 & -a_2\mathbf{W}_2^T & \mathbf{0} & \cdots & \mathbf{0} \\
-a_2\mathbf{W}_2 & \mathbf{I} + a_3^2\mathbf{W}_3^T\mathbf{W}_3 & -a_3\mathbf{W}_3^T & \cdots & \mathbf{0} \\
\mathbf{0} & -a_3\mathbf{W}_3 & \mathbf{I} + a_4^2\mathbf{W}_4^T\mathbf{W}_4 & \ddots & \mathbf{0} \\
\vdots & \vdots & \ddots & \ddots & -a_{L-1}\mathbf{W}_{L-1}^T \\
\mathbf{0} & \mathbf{0} & \mathbf{0} & -a_{L-1}\mathbf{W}_{L-1} & \mathbf{I} + a_L^2\mathbf{W}_L^T\mathbf{W}_L
\end{bmatrix}}_{\mathbf{H_z}}
\underbrace{\begin{bmatrix} \mathbf{z}_1 \\ \mathbf{z}_2 \\ \vdots \\ \mathbf{z}_{H-1} \\ \mathbf{z}_H \end{bmatrix}}_{\mathbf{z}}
- \underbrace{\begin{bmatrix} a_1\mathbf{W}_1\mathbf{x} \\ \mathbf{0} \\ \vdots \\ \mathbf{0} \\ a_L\mathbf{W}_L^T\mathbf{y} \end{bmatrix}}_{\mathbf{b}}
$$

$$(15)$$

where the matrix of coefficients corresponds to the Hessian of the energy with respect to the activities $(\mathbf{H_z})_{\ell k} := \partial^2\mathcal{F}/\partial\mathbf{z}_\ell\partial\mathbf{z}_k$. We make the following side remarks about how different training and architecture design choices impact the structure of the activity Hessian:

- In the unsupervised case where $\mathbf{z}_0$ is left free to vary like any other hidden layer, the Hessian gets the additional terms $a_1^2\mathbf{W}_1^T\mathbf{W}_1$ as the first diagonal block, $-a_1\mathbf{W}_1$ as the superdiagonal block (and its transpose as the subdiagonal block), and $\mathbf{b}_1 = \mathbf{0}$.[11] This does not fundamentally change the structure of the Hessian; in fact, in the next section we show that convexity holds for both the unsupervised and supervised cases.

- Turning on biases at each layer such that $\mathcal{F}_\ell = \frac{1}{2}||\mathbf{z}_\ell - a_\ell\mathbf{W}_\ell\mathbf{z}_{\ell-1} - \mathbf{b}_\ell||^2$ does not impact the Hessian and simply makes the constant vector of the linear system more dense: $\mathbf{b} = [a_1\mathbf{W}_1\mathbf{x} + \mathbf{b}_1 - a_2\mathbf{W}_2^T\mathbf{b}_2, \mathbf{b}_2 - a_3\mathbf{W}_3^T\mathbf{b}_3, \ldots, a_L\mathbf{W}_L^T\mathbf{y} + \mathbf{b}_{L-1} - a_L\mathbf{W}_L^T\mathbf{b}_L]^T$.

- Adding an $\ell^2$ norm regulariser to the activities $\frac{1}{2}||\mathbf{z}_\ell||^2$ scales the identity in each diagonal block by 2. This induces a unit shift in the Hessian eigenspectrum such that the minimum eigenvalue is lower bounded at one rather than zero (see §A.2.3), as shown in Fig. A.12.

- Adding "dummy" latents at either end of the network, such that $\mathcal{F}_0 = \frac{1}{2}||\mathbf{x} - \mathbf{z}_0||^2$ or $\mathcal{F}_L = \frac{1}{2}||\mathbf{y} - \mathbf{z}_L||^2$, simply adds one layer to the Hessian with a block diagonal given by $2\mathbf{I}$.

- Compared to fully connected networks, the activity Hessian of convolutional networks is sparser in that (dense) weight matrices are replaced by (sparser) Toeplitz matrices. The activity Hessian of ResNets is derived and discussed in §A.2.4.

We also note that Eq. 15 can be used to provide an alternative proof of the known convergence of PC inference to the feedforward pass [39] $\mathbf{z}^* = \mathbf{H_z}^{-1}\mathbf{b} = f(\mathbf{x}) = a_L\mathbf{W}_L \ldots a_1\mathbf{W}_1\mathbf{x}$ when the output layer is unclamped or free to vary with $\partial^2\mathcal{F}/\partial\mathbf{z}_L^2 = \mathbf{I}$ and $\mathbf{b}_H = \mathbf{0}$.

### A.2.2 Positive definiteness of the activity Hessian

Here we prove that the Hessian of the energy with respect to the activities of arbitrary DLNs (Eq. 5) is positive definite (PD), $\mathbf{H_z} \succ 0$. The result is empirically verified for DLNs in §A.2.3 and also appears to generally hold for nonlinear networks, where we observe small negative Hessian eigenvalues only for very shallow Tanh networks with no skip connections (see Figs. A.7 & A.22).

> **Theorem A.1** (Convexity of the PC inference landscape of DLNs.). *For any DLN parameterised by $\boldsymbol{\theta} := (\mathbf{W}_1, \ldots, \mathbf{W}_L)$ with input and output $(\mathbf{x}, \mathbf{y})$, the activity Hessian of the PC energy (Eq. 1) is positive definite*
>
> $$\mathbf{H_z}(\boldsymbol{\theta}) \succ 0, \qquad (16)$$
>
> *showing that the inference or activity landscape $\mathcal{F}(\mathbf{z})$ is strictly convex.*

To prove this, we will show that the Hessian satifies *Sylvester's criterion*, which states that a Hermitian matrix is PD if all of its leading principal minors (LPMs) are positive, i.e. if the determinant of all its square top-left submatrices is positive [21]. Recall that an $n \times n$ square matrix $\mathbf{A}$ has $n$ LPMs $\mathbf{A}_h$ of size $h \times h$ for $h = 1, \ldots, n$. For a Hermitian matrix, showing that the determinant of all its LPMs is positive is a necessary and sufficient condition to determine whether the matrix is PD ($\mathbf{A} \succ 0$), and this result can be generalised to block matrices.

---

[11]Note that the lack of an identity term in the block diagonal term comes from the fact that the first layer is not directly predicted by any other layer.

We now show that the activity Hessian of arbitrary DLNs (Eq. 5) satisfies Sylvester's criterion. We drop the Hessian subscript $\mathbf{H}$ for brevity of notation. The proof technique lies in a Laplace or cofactor expansion of the LPMs along the last row. This has an intuitive interpretation in that it starts by proving that the inference landscape of one-hidden-layer PCNs is (strictly) convex, and then proceeds by induction to show that adding layers does not change the result.

The activity Hessian has $NH$ LPMs of size $N\ell \times N\ell$ for $\ell = 1, \ldots, H$. Let $[\mathbf{H}]_\ell$ denote the $\ell$th LPM of $\mathbf{H}$, $\Delta_\ell = |[\mathbf{H}]_\ell|$ its determinant, and $\mathbf{D}_\ell$ and $\mathbf{O}_\ell$ the $\ell$th diagonal and off-diagonal blocks of $\mathbf{H}$, respectively. Now note that $\mathbf{H}$ is a block tridiagonal symmetric matrix, as can be clearly seen from Eq. 15. There is a known two-term recurrence relation that can be used to calculate the determinant of such matrices through their LPMs [53]

$$\Delta_\ell = |\mathbf{D}_\ell|\Delta_{\ell-1} - |\mathbf{O}_{\ell-1}|^2 \Delta_{\ell-2}, \quad \ell = 2, \ldots, H \tag{17}$$

with $\Delta_0 = 1$ and $\Delta_1 = |\mathbf{D}_1|$. The first LPM is clearly PD and so its determinant is positive, $\mathbf{D}_1 = \mathbf{I} + a_2^2 \mathbf{W}_2^T \mathbf{W}_2 \succ 0 \implies \Delta_1 > 0$, showing that the inference landscape of one-hidden-layer linear PCNs is strictly convex. For $\ell = 2$, the first term of the recursion (Eq. 17) is positive, since $|\mathbf{D}_2| = |\mathbf{I} + a_3^2 \mathbf{W}_3^T \mathbf{W}_3| > 0$ and, $\Delta_1 > 0$ as we just saw. The second term is negative, but it is strictly less than the positive term, $|a_2 \mathbf{W}_2|^2 < |\mathbf{I} + a_3^2 \mathbf{W}_3^T \mathbf{W}_3||\mathbf{I} + a_2^2 \mathbf{W}_2^T \mathbf{W}_2|$ and so $\Delta_2 > 0$. Hence, the activity landscape of 2-hidden-layer linear PCNs remains convex. The same holds for three hidden layers where $|\mathbf{O}_2|\Delta_1 < |\mathbf{D}_3|\Delta_2 \implies \Delta_3 > 0$.

We can keep iterating this argument, showing by induction that the inference landscape is (strictly) convex for arbitrary DLNs. More formally, the positive term of the recurrence relation is always strictly greater than the negative term,

$$|\mathbf{D}_\ell|\Delta_{\ell-1} > 0 \tag{18}$$

$$|\mathbf{D}_\ell|\Delta_{\ell-1} > |\mathbf{O}_{\ell-1}|^2 \Delta_{\ell-2} \tag{19}$$

and so $\Delta_\ell > 0$ and $\mathbf{H} \succ 0$ for all $\ell$. Convexity holds for the unsupervised case, where the activity Hessian is now positive *semidefinite* since the term $a_1^2 \mathbf{W}_1^T \mathbf{W}_1$ is introduced (see §A.2.1). The result can also be extended to any other linear layer transformation $\mathbf{B}_\ell$ including ResNets where $\mathbf{B}_\ell = \mathbf{I} + \mathbf{W}_\ell$.

### A.2.3 Random matrix theory of the activity Hessian

Here we analyse the Hessian of the energy with respect to the activities of DLNs (Eq. 5) using random matrix theory (RMT). This analysis follows a line of work using RMT to study the Hessian of neural networks, specifically the Hessian of the loss with respect to the parameters [7, 45, 13, 32, 2]. We note that the structure of the activity Hessian is much simpler than the weight or parameter Hessian, in that for linear networks the former is positive definite (Theorem A.1, §A.2.2), while for the latter this is only true for one hidden layer [22].

In what follows, we recall from §2.2 that the PC energy (Eq. 1) has layer-wise scalings $a_\ell$ for all $\ell$, and the weights are assumed to be drawn from a zero-mean Gaussian $(\mathbf{W}_\ell)_{ij} \sim \mathcal{N}(0, b_\ell)$ with variance set by $b_\ell$.

**Hessian decomposition.** The activity Hessian (Eq. 5) is a challenging matrix to study theoretically as its entries are not i.i.d. even at initialization due to the off-diagonal couplings between layers. However, we can decompose the matrix into its diagonal and off-diagonal components:

$$\mathbf{H_z} = \mathbf{D} + \mathbf{O} \tag{20}$$

with $\mathbf{D} := \text{diag}(\mathbf{I} + a_2^2 \mathbf{W}_2^T \mathbf{W}_2, \ldots, \mathbf{I} + a_L^2 \mathbf{W}_L^T \mathbf{W}_L)$ and $\mathbf{O} := \text{offdiag}(-a_2 \mathbf{W}_2, \ldots, -a_{L-1} \mathbf{W}_{L-1})$, where the off-diagonal part can be seen as a perturbation. Since these matrices are on their own i.i.d. at initialisation, we can use standard RMT results to analyse their respective eigenvalue distributions in the regime of large width $N$ and depth $H$ we are interested in. We will then use these results to gain some qualitative insights into the overall spectrum of $\mathbf{H_z}$.

**Analysis of D.** As a block diagonal matrix, the eigenvales of $\mathbf{D}$ are given by those of its blocks $\mathbf{D}_\ell = \mathbf{I} + a_{\ell+1}^2 \mathbf{W}_{\ell+1}^T \mathbf{W}_{\ell+1} \in \mathbb{R}^{N \times N}$ for $\ell = 1, \ldots, H$. Note that the size of each block depends only on the network width $N$. It is easy to see that each block is a positively shifted Wishart matrix. As $N \to \infty$, the eigenspectrum of such matrices converges to the well-known Marčhenko-Pastur (MP) distribution [35] if properly normalised such that $a_{\ell+1}^2 \mathbf{W}_{\ell+1}^T \mathbf{W}_{\ell+1} \sim \mathcal{O}(1/N)$.

As shown in Figs. A.1-A.2, this normalisation can be achieved in two distinct but equivalent ways: (i) by initialising from a standard Gaussian with $b_\ell = 1$ and setting the layer scaling to $a_\ell = 1/\sqrt{N}$, or (ii) by setting $a_\ell = 1$ and $b_\ell = 1/N$ as done by standard initialisations [30, 11, 18]. In either case, in the infinite-width limit the eigenvalues of each diagonal block will converge to a unit-shifted MP density with extremes

$$\lim_{N\to\infty} \lambda_\pm(\mathbf{D}_\ell) = 1 + (1 \pm \sqrt{N/N})^2 \quad (21)$$

$$= \{1, 5\}. \quad (22)$$

While the spectrum of $\mathbf{D}$ will be a combination of these independent MP densities, its extremes will be the same of $\mathbf{D}_\ell$ since all of the blocks are i.i.d. and grow at the same rate as $N \to \infty$. This is empirically verified in Figs. A.1-A.2, which also confirm that the spectrum of $\mathbf{D}$ is only affected by the width and not the depth.

**Analysis of O.** The off-diagonal component of the Hessian $\mathbf{O}$ is a sparse Wigner matrix whose size depends on both the width and the depth and so the correct limit should take both $N, H \to \infty$ at some constant ratio. Note that the sparsity of $\mathbf{O}$ grows much faster with the depth. Because sparse Wigner matrices are poorly understood and still an active area of research [62], we make the simplifying assumption that $\mathbf{O}$ is dense.

If properly normalised as above, we know that in the limit the eigenspectrum of dense Wigner matrices converges the classical Wigner semicircle distribution [67] with extremes

$$\lim_{H/N\to\infty} \lambda_\pm(\mathbf{O}) = \pm 2. \quad (23)$$

We find that the empirical eigenspectrum of $\mathbf{O}$ is slightly broader than the semicircle and, as expected, is affected by both the width and the depth (Figs. A.3-A.4).

**Analysis of $\mathbf{H_z}$.** Given the above asymptotic results on $\mathbf{D}$ and $\mathbf{O}$, we can use Weyl's inequalities [65] to lower and upper bound the minimum and maximum eigenvalues (and so the condition number) of the overall Hessian at initialisation: $\lambda_{\max}(\mathbf{D} + \mathbf{O}) \leq \lambda_{\max}(\mathbf{D}) + \lambda_{\max}(\mathbf{O})$ and $\lambda_{\min}(\mathbf{D}+\mathbf{O}) \geq \lambda_{\min}(\mathbf{D}) + \lambda_{\min}(\mathbf{O})$. The upper bound ($\tilde{\lambda}_{\max} = 7$) appears tight, as shown in Figs. A.5-A.7. However, the lower bound predicts a negative minimum eigenvalue ($\tilde{\lambda}_{\min} = -1$), which is not possible since the Hessian is positive definite as we proved in §A.2.2.

Nevertheless, we can still gain some insights into the interaction between $\mathbf{D}$ and $\mathbf{O}$ by looking at the empirical eigenspectrum of $\mathbf{H_z}$. In particular,

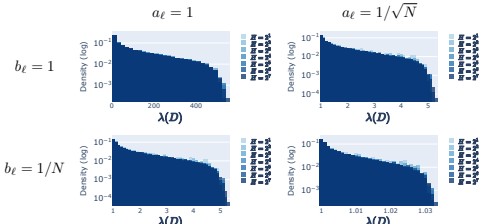

Figure A.1: **Empirical eigenspectra of D at initialisation, holding the network width constant ($N = 128$) and varying the depth $H$.** $a_\ell$ indicates the premultiplier at each network layer (Eq. 1), while $b_\ell$ is the variance of Gaussian initialisation, with $a_\ell = 1$ and $b_\ell = 1/N$ corresponding to the "standard parameterisation" (SP).

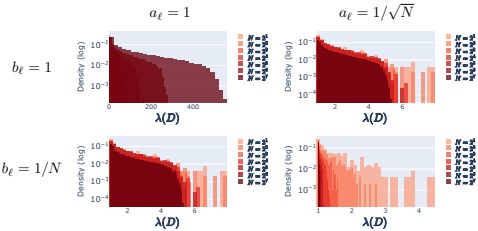

Figure A.2: **Empirical eigenspectra of D at initialisation, holding the network depth constant ($H = 128$) and varying the width $N$.**

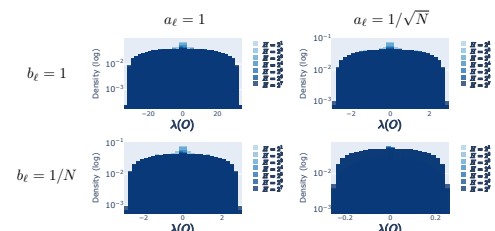

Figure A.3: **Empirical eigenspectra of O at initialisation, holding the network width constant ($N = 128$) and varying the depth $H$.**

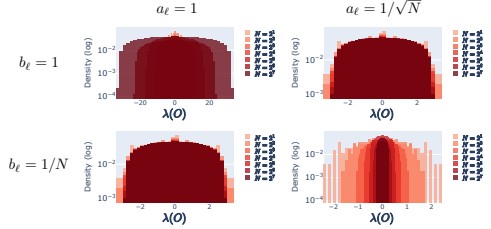

Figure A.4: **Empirical eigenspectra of O at initialisation, holding the network depth constant ($H = 128$) and varying the width $N$.**

we observe that the maximum and especially the minimum eigenvalue of the Hessian scale with the network depth (Figs. A.7 & A.22), thus driving the growth of the condition number.

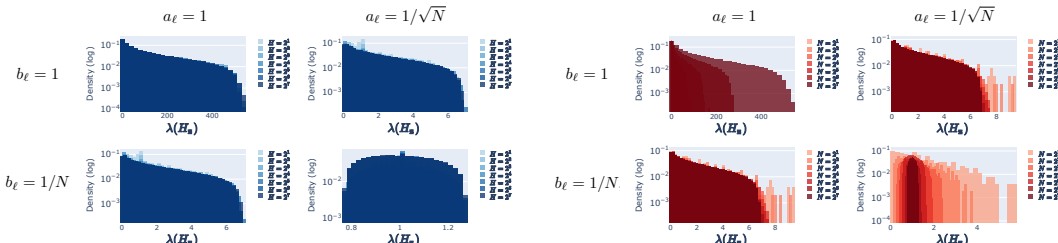

Figure A.5: **Empirical eigenspectra of H at initialisation, holding the network width constant ($N = 128$) and varying the depth $H$.**

Figure A.6: **Empirical eigenspectra of H at initialisation, holding the network depth constant ($H = 128$) and varying the width $N$.**

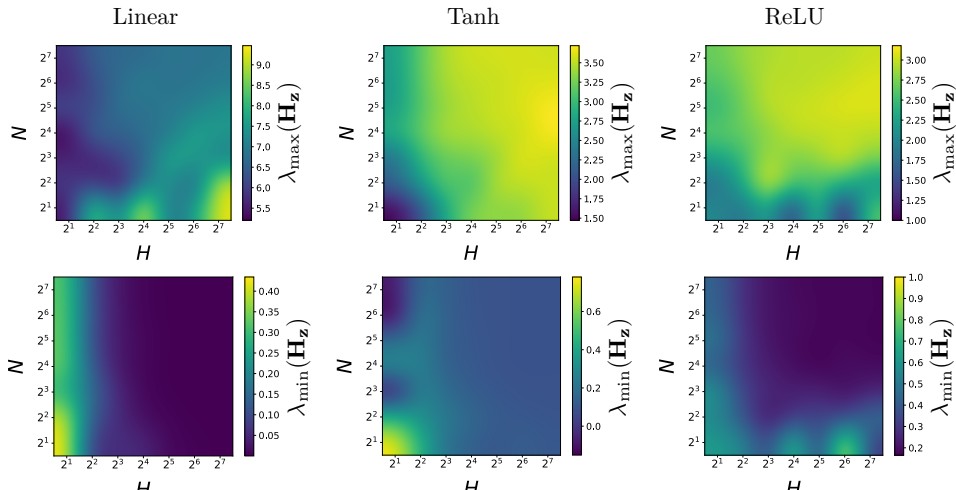

Figure A.7: **Maximum and minimum eigenvalues of $\mathbf{H_z}$ at initialisation as a function of network width $N$ and depth $L$.**

### A.2.4 Activity Hessian of linear ResNets

Here we derive the activity Hessian for linear ResNets [19], extending the derivation in §A.2.1 for DLNs. Following the Depth-$\mu$P parameterisation [72, 3], we consider ResNets with identity skip connections at every layer except from the input and to the output. The PC energy for such ResNets is given by

$$\mathcal{F}_{\text{1-skip}} = \frac{1}{2}||\boldsymbol{\epsilon}_L||^2 + \frac{1}{2}||\boldsymbol{\epsilon}_1||^2 + \sum_{\ell=2}^{H} \frac{1}{2}||\mathbf{z}_\ell - a_\ell \mathbf{W}_\ell \mathbf{z}_{\ell-1} - \underbrace{\mathbf{z}_{\ell-1}}_{\text{1-skip}}||^2, \tag{24}$$

where recall that $\epsilon_\ell = \mathbf{z}_\ell - a_\ell \mathbf{W}_\ell \mathbf{z}_{\ell-1}$ and $\mathbf{z}_0 := \mathbf{x}$, $\mathbf{z}_L := \mathbf{y}$. We refer to this model as "1-skip" since the residual is added to every layer. Its activity Hessian is given by

$$\mathbf{H_z}^{\text{1-skip}} := \frac{\partial^2 \mathcal{F}_{\text{1-skip}}}{\partial \mathbf{z}_\ell \partial \mathbf{z}_k} = \begin{cases} 2\mathbf{I} + a_{\ell+1}^2 \mathbf{W}_{\ell+1}^T \mathbf{W}_{\ell+1} + a_{\ell+1}(\mathbf{W}_{\ell+1}^T + \mathbf{W}_{\ell+1}), & \ell = k \neq H \\ \mathbf{I} + a_{\ell+1}^2 \mathbf{W}_{\ell+1}^T \mathbf{W}_{\ell+1}, & \ell = k = H \\ -a_{k+1}\mathbf{W}_{k+1} - \mathbf{I}, & \ell - k = 1 \\ -a_{\ell+1}\mathbf{W}_{\ell+1}^T - \mathbf{I}, & \ell - k = -1 \\ \mathbf{0}, & \text{else} \end{cases} \quad (25)$$

We find that this Hessian is much more ill-conditioned (Fig. A.22) than that of networks without skips (Fig. 2), across different parameterisations (Fig. 4). We note that one can extend these results to $n$-skip linear ResNets with energy

$$\mathcal{F}_{n\text{-skip}} = \frac{1}{2}||\epsilon_L||^2 + \sum_{\ell=1}^{n} \frac{1}{2}||\epsilon_\ell||^2 + \sum_{\ell=n+1}^{H} \frac{1}{2}||\mathbf{z}_\ell - a_\ell \mathbf{W}_\ell \mathbf{z}_{\ell-1} - \underbrace{\mathbf{z}_{\ell-n}}_{n\text{-skip}}||^2 \quad (26)$$

or indeed arbitrary computational graphs [55]. It could be interesting to investigate whether there exist architectures with better conditioning of the inference landscape that do not sacrifice the stability of the forward pass (see §4, Fig. 4).

### A.2.5 Extension to other energy-based algorithms

Here we include a preliminary investigation of the inference dynamics of other energy-based local learning algorithms. As an example, we consider equilibrium propagation (EP) [59], whose energy for a DLN is given by

$$E = \frac{1}{2}||\mathbf{z}_\ell||^2 - \sum_{\ell=1}^{L} \mathbf{z}_\ell^T \mathbf{W}_\ell \mathbf{z}_{\ell-1} + \frac{\beta}{2}||\mathbf{y} - \mathbf{z}_L||^2, \quad (27)$$

where $\mathbf{z}_0 := \mathbf{x}$ for supervised learning (as for PC), and it is also standard to include an $\ell^2$ regulariser on the activities. Unlike PC, EP has two inference phases: a *free* phase where the output layer $\mathbf{z}_L$ is free to vary like any other hidden layer with $\beta = 0$; and a *clamped* or *nudged* phase where the output is fixed to some target $\mathbf{y}$ with $\beta > 0$. The activity gradient and Hessian of the EP energy (Eq. 27) are given by

$$\frac{\partial E}{\partial \mathbf{z}_\ell} = \begin{cases} \mathbf{z}_\ell - \mathbf{W}_\ell \mathbf{z}_{\ell-1} - \mathbf{z}_{\ell+1}^T \mathbf{W}_{\ell+1}, & \ell \neq L \\ \mathbf{z}_\ell - \mathbf{W}_\ell \mathbf{z}_{\ell-1} - \beta(\mathbf{y} - \mathbf{z}_\ell), & \ell = L \end{cases} \quad (28)$$

and

$$\mathbf{H_z} := \frac{\partial^2 E}{\partial \mathbf{z}_\ell \partial \mathbf{z}_k} = \begin{cases} \mathbf{I}, & \ell = k \neq L \\ \mathbf{I} + \beta, & \ell = k = L \\ -\mathbf{W}_{\ell+1}, & \ell - k = 1 \\ -\mathbf{W}_{k+1}^T, & \ell - k = -1 \\ \mathbf{0}, & \text{else} \end{cases} \quad (29)$$

where we abuse notation by denoting the Hessian in the same way as that of the PC energy. We observe that the off-diagonal blocks are equal to those of the PC activity Hessian (Eq. 5). Similar to PC, one can also rewrite the EP activity gradient (Eq. 28) as a linear system

$$\nabla_{\mathbf{z}} E = \underbrace{\begin{bmatrix} \mathbf{I} & -\mathbf{W}_2^T & \mathbf{0} & \dots & \mathbf{0} \\ -\mathbf{W}_2 & \mathbf{I} & -\mathbf{W}_3^T & \dots & \mathbf{0} \\ \mathbf{0} & -\mathbf{W}_3 & \mathbf{I} & \ddots & \mathbf{0} \\ \vdots & \vdots & \ddots & \ddots & -\mathbf{W}_L^T \\ \mathbf{0} & \mathbf{0} & \mathbf{0} & -\mathbf{W}_L & \mathbf{I} + \beta \end{bmatrix}}_{\mathbf{H_z}} \underbrace{\begin{bmatrix} \mathbf{z}_1 \\ \mathbf{z}_2 \\ \vdots \\ \mathbf{z}_{L-1} \\ \mathbf{z}_L \end{bmatrix}}_{\mathbf{z}} - \underbrace{\begin{bmatrix} \mathbf{W}_1 \mathbf{x} \\ \mathbf{0} \\ \vdots \\ \mathbf{0} \\ \beta \mathbf{y} \end{bmatrix}}_{\mathbf{b}} \quad (30)$$

with solution $\mathbf{z}^* = \mathbf{H_z}^{-1} \mathbf{b}$. Interestingly, unlike for PC, the EP inference landscape is not necessarily convex, which can be easily seen for a shallow 2-layer scalar network where $\exists \lambda(\mathbf{H_z}(w_2 > 1)) < 0$. This is always true without the activity regulariser, in which case the identity in each diagonal block vanishes.

### A.2.6 Limit convergence of $\mu$PC to BP (Thm. 1)

Here we provide a simple proof of Theorem 1. Consider a slight generalisation to linear ResNets (Eq. 24) of the PC energy at the inference equilibrium derived by [22] for DLNs:

$$\mathcal{F}(\mathbf{z}^*) = \frac{1}{2B} \sum_{i=1}^{B} \mathbf{r}_i^T \mathbf{S}^{-1} \mathbf{r}_i, \tag{31}$$

$$\text{where} \quad \mathbf{S} = \mathbf{I}_{d_y} + a_L^2 \mathbf{W}_L \mathbf{W}_L^T + \sum_{\ell=2}^{H} \left( a_L \mathbf{W}_L \prod_{\ell}^{H} \mathbf{I} + a_\ell \mathbf{W}_\ell \right) \left( a_L \mathbf{W}_L \prod_{\ell}^{H} \mathbf{I} + a_\ell \mathbf{W}_\ell \right)^T \tag{32}$$

and the residual error is $\mathbf{r}_i = \mathbf{y}_i - a_L \mathbf{W}_L \left( \prod_{\ell=2}^{H} \mathbf{I} + a_\ell \mathbf{W}_\ell \right) a_1 \mathbf{W}_1 \mathbf{x}_i$. $B$ can stand for the batch or dataset size. Note that Eq. 31 is an MSE loss with a weight-dependent rescaling (Eq. 32). Now, we know that, for Depth-$\mu$P, the forward pass of this model has $\mathcal{O}_{N,H}(1)$ preactivations at initialisation and so the residual will also be of order 1. Note that, by contrast, for SP ($a_\ell = 1$ for all $\ell$ and $b_\ell = 1/N_{\ell-1}$) the preactivations explode with the depth (Fig. A.30).

The key question, then, is what happens to the rescaling $\mathbf{S}$ in the limit of large depth and width. Recall that for $\mu$PC, $a_L = 1/N$ and $a_\ell = 1/\sqrt{NL}$ for $\ell = 2, \ldots, H$ (see Table 1). Because the output weights factor in every term of the rescaling $\mathbf{S}$ except for the identity, these terms will all vanish at a $1/N$ rate as $N \to \infty$, i.e. $\mathbf{W}_L \mathbf{W}_L^T / N^2 \sim \mathcal{O}(1/N)$. The depth, on the other hand, scales the number of terms in $\mathbf{S}$. Therefore, the width will have to grow with the depth at some constant ratio $L/N$—which can be thought of as the aspect ratio of the network [50]—to make the contribution of each term as small as possible. In the limit of this ratio $r \to 0$, the energy rescaling (Eq. 32) approaches the identity $\mathbf{S} = \mathbf{I}$, the equilibrated energy converges to the MSE $\mathcal{F}_{\mu\text{PC}}(\mathbf{z}^*, \boldsymbol{\theta}) = \mathcal{L}_{\mu\text{P}}(\boldsymbol{\theta})$, and so PC computes the same gradients as BP.

## A.3 Additional experiments

### A.3.1 Ill-conditioning with training

For the setting in Fig. 3, we also ran experiments with Adam as inference algorithm and ResNets with standard GD. All the results were tuned for the weight learning rate (see §A.4 for more details). We found that Adam led to more ill-conditioned inference landscapes associated with significantly lower and more unstable performance than GD (Figs. 3 & A.23).

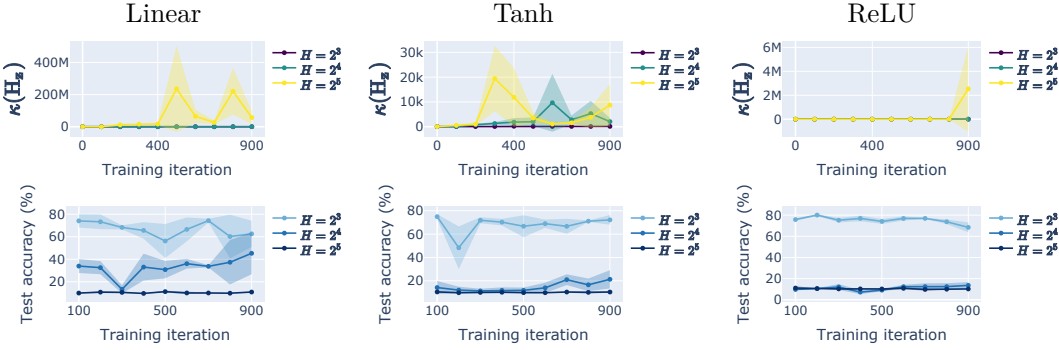

Figure A.8: **Same results as Fig. 3 with Adam as inference algorithm (MNIST).**

Interestingly, while skip connections induced much more extreme ill-conditioning (Fig. A.22), performance was equal to, and sometimes significantly better than, networks without skips (Figs. A.9 & A.25), suggesting a complex relationship between trainability and the geometry of the inference landscape which we return to in §A.3.6.

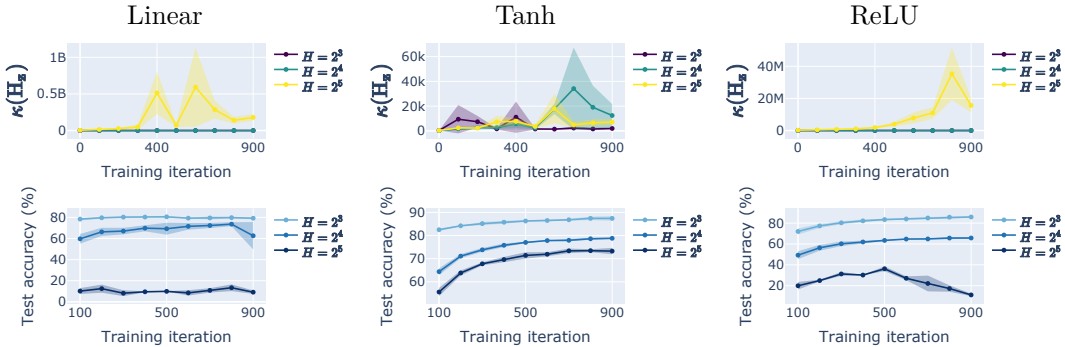

Figure A.9: **Same results as Fig. 3 with skip connections (MNIST).**

### A.3.2 Activity initialisations

Here we present some additional results on the initialisation of the activities of PCNs. All experiments used fully connected ResNets, GD as activity optimiser, and as many inference steps as the number of hidden layers. For intuition, we start with linear scalar PCNs or chains. First, we verify that the ill-conditioning of the inference landscape (§3.1) causes the activities to barely move during inference, and increasing the activity learning rate leads to divergence for both forward and random initialisation (Fig. A.10). Similar results are observed for $\mu$PC (see Fig. A.35).

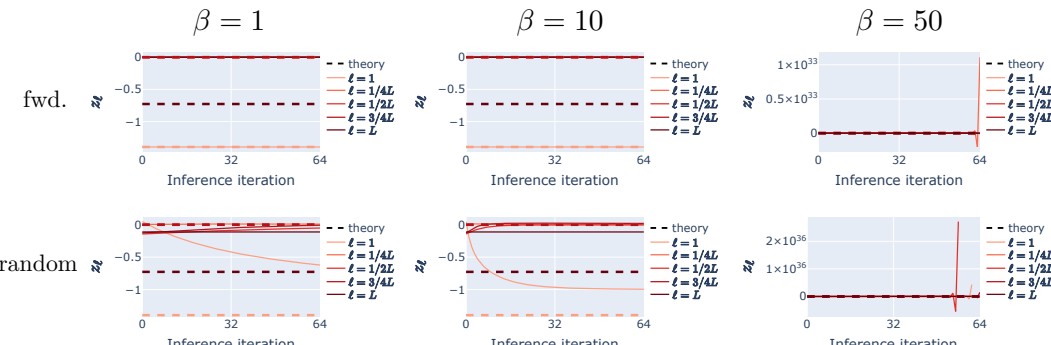

Figure A.10: **Ill-conditioning of the inference landscape prevents convergence to the analytical solution regardless of initialisation.** For different initialisations (forward and random) and activity learning rates $\beta$, we plot the activities of a 64-layer scalar PCN over inference at the start of training. The theoretical activities were computed using Eq. 4. The task was a simple toy regression with $y = -x + \epsilon$ with $x \sim \mathcal{N}(1, 1)$ and $\epsilon \sim \mathcal{N}(0, 0.5)$. A standard Gaussian was used for random initialisation, $z_\ell \sim \mathcal{N}(0, 1)$. Results were similar across different random seeds.

For wide linear PCNs with forward initialisation, we find similar results except that $\mu$PC seems to initialise the activities close to the analytical solution (Fig. A.11). The same pattern of results is observed for nonlinear networks (Fig. A.36), although note that in this case we do have an analytical solution. These results might suggest that one does not need to perform many inference steps to achieve good performance with $\mu$PC. However, we found that one inference step led to worse performance (including as a function of depth) (Figs. A.14 & A.27) compared to as many steps as number of hidden layers (Figs. A.16 & A.18).

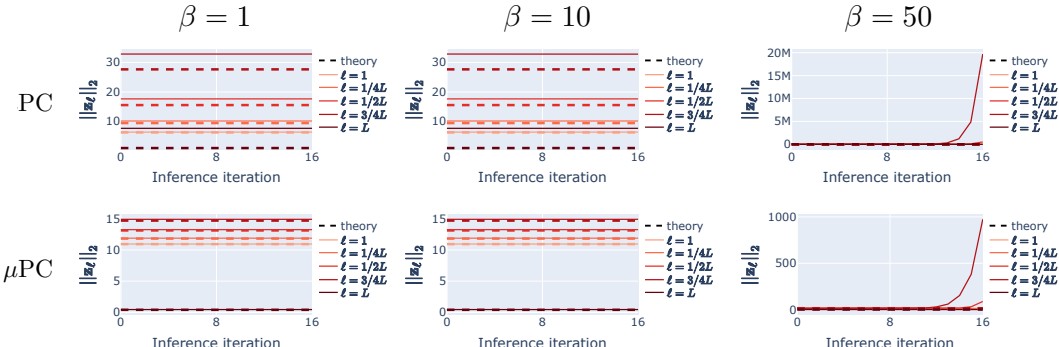

Figure A.11: **The forward pass of $\mu$PC seems to initialise the activities close to the analytical solution (Eq. 4).** Similar to Fig. A.10, we plot the $\ell^2$ norm of the activities over inference of 16-layer linear PCNs ($N = 128$) at the start of training (MNIST). Again, results were similar across different random initialisations.

### A.3.3 Activity decay

In §4, we discussed how it seems impossible to achieve good conditioning of the inference landscape without making the forward pass unstable (e.g. by zeroing out the weights). We identified one way of inducing relative well-conditionness at initialisation without affecting the forward pass, namely adding an $\ell^2$ norm regulariser on the activities $\frac{\alpha}{2} \sum_\ell^H ||\mathbf{z}_\ell||^2$ with $\alpha = 1$. This effectively induces a unit shift in the Hessian spectrum and bounds the minimum eigenvalue at one rather than zero (see §A.2.3). However, we find that PCNs with *any degree of activity regularisation* $\alpha$ are untrainable (Fig. A.12).

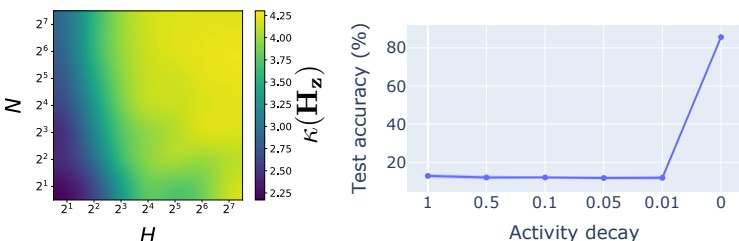

Figure A.12: **Activity decay induces well-conditioned inference at the cost of performance.** *Left*: Same plot as Fig. 2 with an added activity regulariser $\frac{\alpha}{2}||\mathbf{z}_\ell||^2$ with $\alpha = 1$. *Right*: Maximum test accuracy on MNIST achieved by a linear PCN with $N = 128$ and $H = 8$ over activity regularisers of varying strength $\alpha$. Solid lines and (barely visible) shaded regions indicate the mean and standard deviation across 3 random seeds, respectively.

### A.3.4 Orthogonal initialisation

As mentioned in §5, in addition to $\mu$PC we also tested PCNs with orthogonal initialisation as a parameterisation ensuring stable forward passes at initialisation for some activation functions (§4; Fig. A.30). We found that this initialisation was not as effective as $\mu$PC (Figs. A.13 & A.26), likely due to loss of orthogonality of the weights during training. Adding an orthogonal regulariser could help, but at the cost of an extra hyperparameter to tune. We also find that, except for linear networks, the ill-conditioning of the inference landscape still grows and spikes during training, similar to other parameterisations (e.g. Fig. 3).

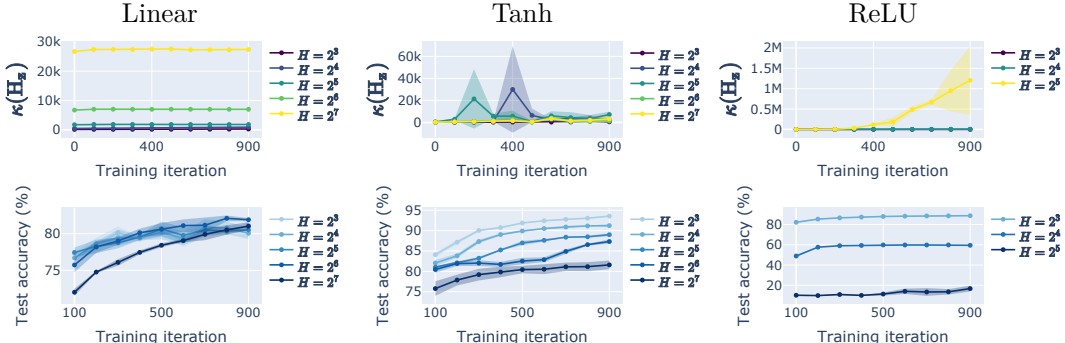

Figure A.13: **Test accuracies in Fig. 1 for orthogonal initialisation.** Note that performance is expected to drop for ReLU networks which cannot have stable forward passes with orthogonal weights (Fig. A.30). We also plot the condition number of the activity Hessian over training.

### A.3.5  $\mu$PC with one inference step

All the experiments with $\mu$PC (e.g. Fig. 1) used as many inference steps as hidden layers. Motivated by the results of §A.3.2 showing that the forward pass of $\mu$PC seems to initialise the activities close to the analytical solution for DLNs (Eq. 4), we also performed experiments with a single inference step. We found that this led a degradation in performance not only at initialisation but also as a function of depth (Figs. A.14 & A.27), suggesting that some number of steps is still necessary despite $\mu$PC appearing to initialise the activities close to the inference solution (Fig. A.11). Similar to other parameterisations, we find that the ill-conditioning of the inference landscape grows and spikes during training.

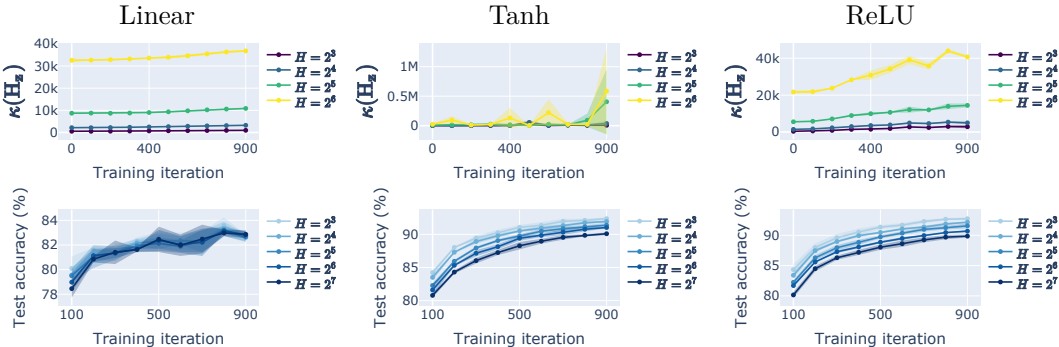

Figure A.14: $\mu$**PC test accuracies in Fig. 1 with one inference step.** We also plot the condition number of the activity Hessian during training.

### A.3.6  Is inference convergence sufficient for good generalisation?

Our analysis of the conditioning of the inference landscape (§3.1) could be argued to rely on the assumption that converging to a solution of the inference dynamics is beneficial for learning and ultimately performance. This question has yet to be resolved, with some works showing both theoretical and empirical benefits for learning close to the inference equilibrium [61, 22], while others argue to take only one step [57]. As discussed in §7, our results suggest that convergence close to a solution is necessary for successful training (or monotonic decrease of the loss), which for brevity we will refer to as "trainability". In particular, $\mu$PC seems to the activities much closer to the analytical solution (Eq. 4) than the SP (§A.3.2), and training $\mu$PC with one inference step leads to worse performance (e.g. Fig. A.14) than with as many as hidden layers (e.g. Fig. 1).

Here we report another experiment that speaks to this question and in particular suggests that *while inference convergence is necessary for trainability, it is insufficient for good generalisation*, at least

for standard PC. Training linear ResNets of varying depth on MNIST with "perfect inference" (using Eq. 4), we observe that even the deepest ($H = 32$) networks now become trainable with standard PC in the sense that the training and test losses decrease monotonically (Fig. A.15). However, the starting point of the test losses substantially increases with the depth, and the test accuracies of the deepest networks remain at chance level. These results do not contradict our analysis but highlight the important distinction between trainability and generalisation. Our analysis addresses the former, while the latter is beyond the scope of this work.

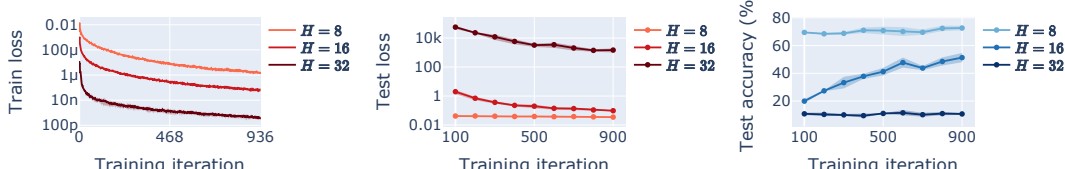

Figure A.15: **Train and test metrics of standard PCNs of varying depth trained with analytical inference (Eq. 4).** We plot the training loss, test loss and test accuracy of ResNets ($N = 128$) trained with standard PC on MNIST by solving for inference analytically (using Eq. 4). All experiments used Adam as optimiser with learning rate $\eta = 1e^{-3}$. Solid lines and shaded regions represent the mean and standard deviation across 3 random initialisations.

## A.4 Experimental details

Code to reproduce all the experiments is available at https://github.com/thebuckleylab/jpc/experiments/mupc_paper. We always used no biases, batch size $B = 64$, Adam as parameter optimiser, and GD as inference optimiser (with the exception of Figs. A.8 & A.24). For the SP, all networks used Kaiming Uniform $(\mathbf{W}_\ell)_{ij} \sim \mathcal{U}(-1/N_{\ell-1}, 1/N_\ell)$ as the standard (PyTorch) initialisation used to train PCNs.

$\mu$**PC experiments (e.g. Fig. 1).** For the test accuracies in Figs. 1 & A.16, we trained fully connected ResNets (Eq. 24) to classify MNIST with standard PC, $\mu$PC and BP with Depth-$\mu$P. To ensure fair comparison, BP with Depth-$\mu$P employed the same scalings as $\mu$PC. All networks had width $N = 512$ and always used as many GD inference iterations as the number of hidden layers $H \in \{2^i\}_{i=3}^7$. To save compute, we trained only for one epoch and evaluated the test accuracy every 300 iterations. For $\mu$PC, we selected runs based on the best results from the depth transfer (see **Hyperparameter transfer** below). For standard PC, we conducted the same grid search over the weight and activity learning rates as used for $\mu$PC. For BP, we performed a sweep over learning rates $\eta \in \{1e^0, 5e^{-1}, 1e^{-1}, 5e^{-2}, 1e^{-2}, 5e^{-3}, 1e^{-3}, 5e^{-4}, 1e-4\}$ at depth $H = 8$, and transferred the optimal value to the deepest ($H = 128$) networks presented.

Fig. A.20 shows similar results for $\mu$PC based on the width transfer results. Fig. A.17 was obtained by extending the training of the 128 ReLU networks in Fig. 1 to 5 epochs. Figs. A.14 & A.27 were obtained with the same setup as Fig. 1 by running $\mu$PC for a single inference step. As noted in §5, the results on Fashion-MNIST (Fig. A.18) were obtained with depth transfer by tuning 8-layer networks and transferring the optimal learning rates to 128 layers.

**Hessian condition number at initialisation (e.g. Fig. 2).** For different activation functions (Fig. 2), architectures (Fig. A.22) and parameterisations (Fig. 4), we computed the condition number of the activity Hessian (Eq. 5) at initialisation over widths and depths $N, H \in \{2^i\}_{i=1}^7$. This was the maximum range we could achieve to compute the full Hessian matrix given our memory resources. No biases were used since these do not affect the Hessian as explained in §A.2.1. Results did not differ significantly across different seeds or input and output data dimensions, as predicted from the structure of the activity Hessian (Eq. 5).

For the landscape insets of Fig. 2, the energy landscape was sampled around the linear solution of the activities (Eq. 4) along the maximum and minimum eigenvectors of the Hessian $\mathcal{F}(\mathbf{z}^* + \alpha\hat{\mathbf{v}}_{\min} + \beta\hat{\mathbf{v}}_{\min})$, with domain $\alpha, \beta \in [-2, 2]$ and $30 \times 30$ resolution.

**Hessian condition number over training (e.g. Fig. 3).** For different activations (e.g. Fig. 3), architectures (e.g. Fig. A.9), algorithms (e.g. Fig. A.8) and parameterisations (e.g. Fig. A.13), we trained networks of width $N = 128$ and hidden layers $H \in \{8, 16, 32\}$ to perform classification on MNIST and Fashion-MNIST. This set of widths and depths was chosen to allow for tractable computation of the full activity Hessian (Eq. 5). Training was stopped after one epoch to illustrate the phenomenon of ill-conditioning. All experiments used weight learning rate $\eta = 1e^{-3}$ and performed a grid search over activity learning rates $\beta \in \{5e^{-1}, 1e^{-1}, 5e^{-2}\}$. A maximum number of $T = 500$ steps was used, and inference was stopped when the norm of the activity gradients reached some tolerance.

**Hyperparameter transfer (e.g. Fig. 5).** For the ResNets trained on MNIST with $\mu$PC (e.g. Fig. 1), we performed a 2D grid search over the following learning rates: $\eta \in \{5e^{-1}, 1e^{-1}, 5e^{-2}, 1e^{-2}\}$ for the weights, and $\beta \in \{1e^3, 5e^2, 1e^2, 5e^1, 1e^1, 5e^0, 1e^0, 5e^{-1}, 1e^{-1}, 5e^{-2}, 1e^{-2}\}$ for the activities. We trained only for one epoch, in part to save compute and in part based on the results of [3, Fig. B.3] showing that the optimal learning rate could be decided after just 3 epochs on CIFAR-10. The number of (GD) inference iterations was always the same as the number of hidden layers. For the width transfer results, we trained networks of 8 hidden layers and widths $N \in \{2^i\}_{i=6}^{10}$, while for the depth transfer we fixed the width to $N = 512$ and varied the depth $H \in \{2^i\}_{i=3}^{7}$. Note that this means that the plots with title $N = 512$ and $H = 8$ in Figs. 5 & A.31-A.32 are the same. The landscape contours were averaged over 3 different random seeds, and the training loss is plotted on a log scale to aid interpretation.

**Loss vs energy ratios (e.g. Fig. 6).** We trained ResNets (Eq. 24) to classify MNIST for one epoch with widths and depths $N, H \in \{2^i\}_{i=1}^{6}$. To replicate the successful setup of Fig. 1, we used the same learning rate for the optimal linear networks trained on MNIST, $\eta = 1e^{-1}$. To verify Theorem 1, at every training step we computed the ratio between the Depth-$\mu$P MSE loss $\mathcal{L}(\boldsymbol{\theta})$ and the equilibrated $\mu$PC energy $\mathcal{F}(\mathbf{z}^*, \boldsymbol{\theta})$ (Eq. 31), where $\mathbf{z}^*$ was computed using Eq. 4. All experiments used the weight learning rate $\eta = 1e^{-4}$. Fig. A.33 shows the same results for the SP, which used a smaller learning rate $\eta = 1e^{-4}$ to avoid divergence at large depth. All the phase diagrams are plotted on a log scale for easier visualisation. Fig. A.34 shows an example of the ratio dynamics of $\mu$PC vs PC for a ResNet with 4 hidden layers and different widths. Results were similar across different random initialisations.

## A.5 Compute resources

The experiments involving $\mu$PC, hyperparameter transfer, and the monitoring of the condition number of the Hessian during training were all run on an NVIDIA RTX A6000. The runtime varied by experiment, with the 128-layer networks trained for multiple epochs (Figs. A.17-A.18) taking several days. All other experiments were run on a CPU and took between one hour and half a day, depending on the specific experiment.

## A.6 Supplementary figures

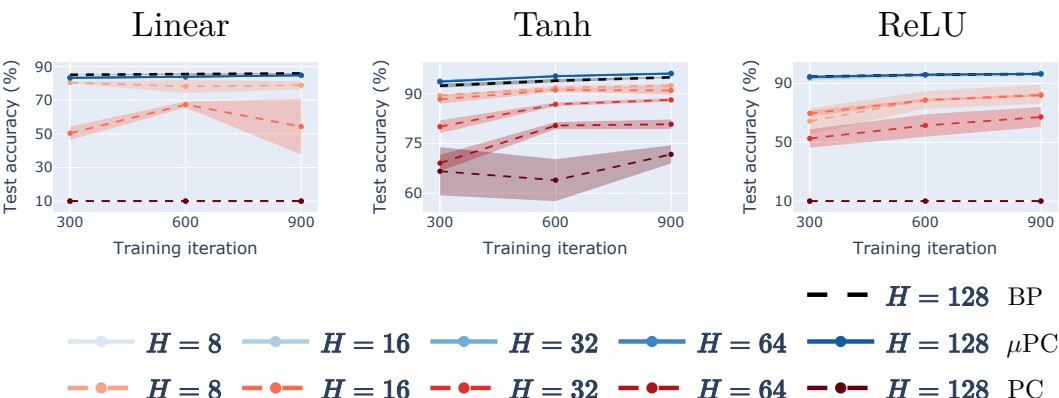

Figure A.16: **Test accuracies in Fig. 1 for different activation functions.** Solid lines and shaded regions indicate the mean and standard deviation across 3 random seeds, respectively. BP represents BP with Depth-$\mu$P.

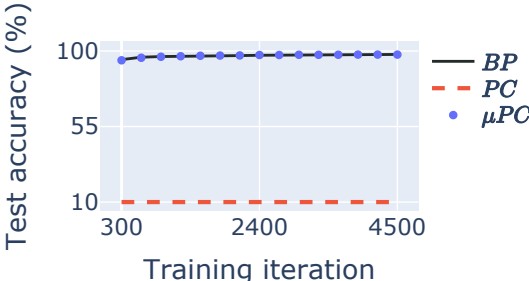

Figure A.17: **128-layer residual ReLU network trained with $\mu$PC on MNIST for 5 epochs.** Solid lines and (barely visible) shaded regions indicate the mean and standard deviation across 5 random seeds, respectively. BP represents BP with Depth-$\mu$P.

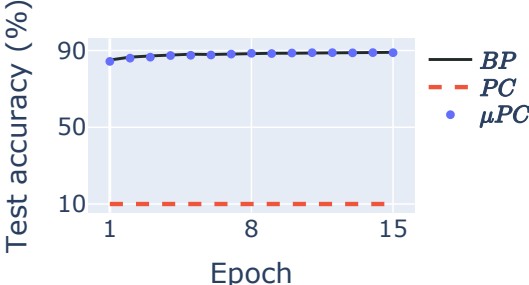

Figure A.18: **128-layer residual ReLU network trained with $\mu$PC on Fashion-MNIST.** Solid lines and (barely visible) shaded regions indicate the mean and standard deviation across 3 random seeds, respectively. BP represents BP with Depth-$\mu$P.

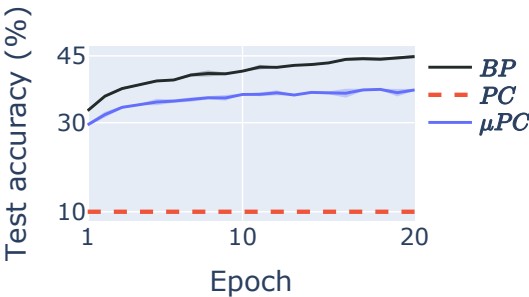

Figure A.19: **128-layer fully connected residual ReLU network trained with $\mu$PC on CIFAR10.** Solid lines and (barely visible) shaded regions indicate the mean and standard deviation across 3 random seeds, respectively. BP represents BP with Depth-$\mu$P. As for other datasets, we see that $\mu$PC remains capable of training such deep networks, although performance slightly lags behind BP. Note that accuracies for all algorithms are far from SOTA because of the fully connected (as opposed to convolutional) architecture used.

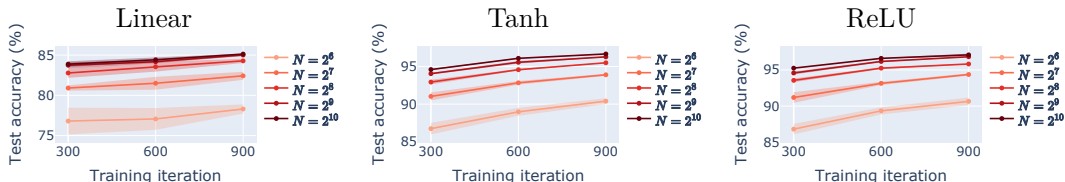

Figure A.20: **Same results as Fig. 1 varying the width $N$ and fixing the depth at $H = 8$, showing that "wider is better" [69, 26].**

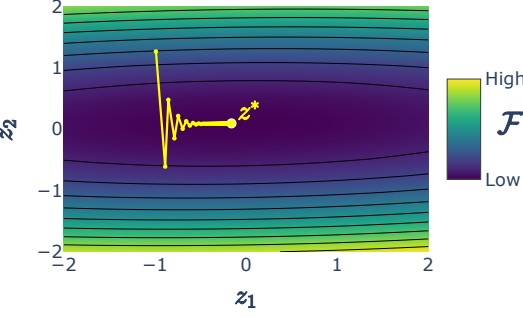

Figure A.21: **Toy illustration of the ill-conditioning of the inference landscape.** Plotted is the activity or inference landscape $\mathcal{F}(z_1, z_2)$ for a toy linear network with two hidden units $f(x) = w_3 w_2 w_1 x$, along with the GD dynamics. One weight was artificially set to a much higher value than the others to induce ill-conditioning.

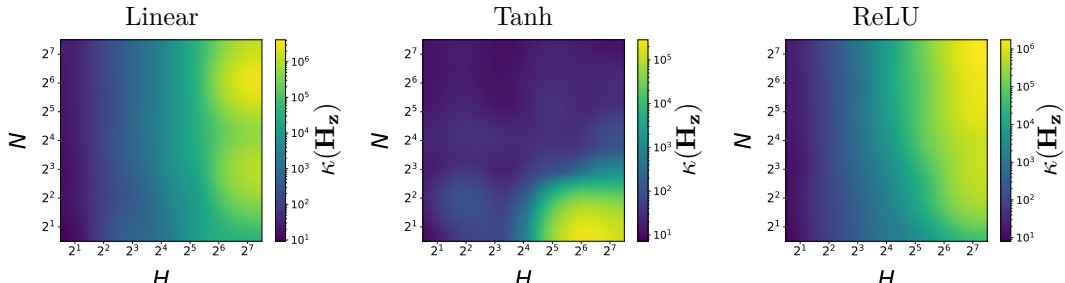

Figure A.22: **Same results as Fig. 2 for the activity Hessian of ResNets (Eq. 25).**

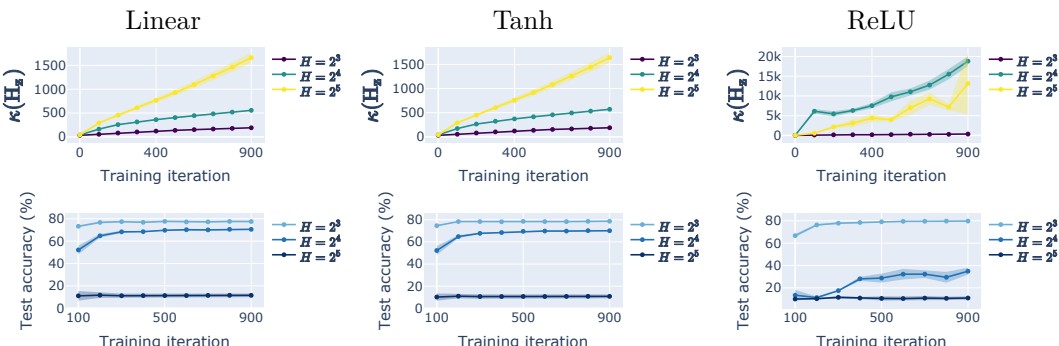

Figure A.23: **Same results as Fig. 3 for Fashion-MNIST.**

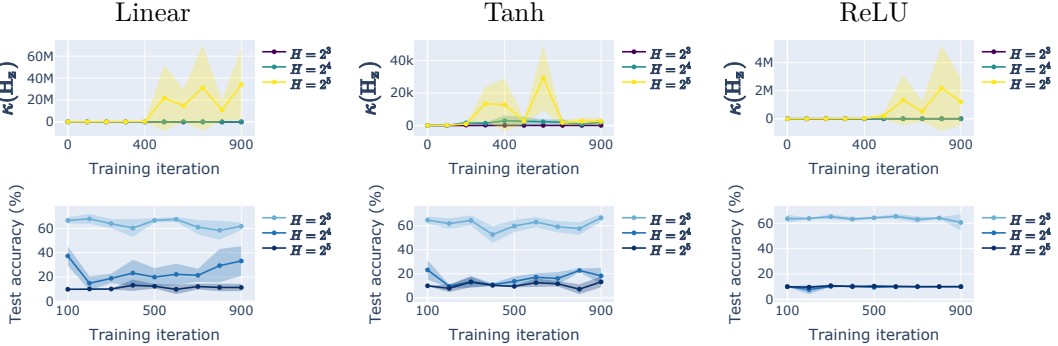

Figure A.24: **Same results as Fig. A.8 for Fashion-MNIST.**

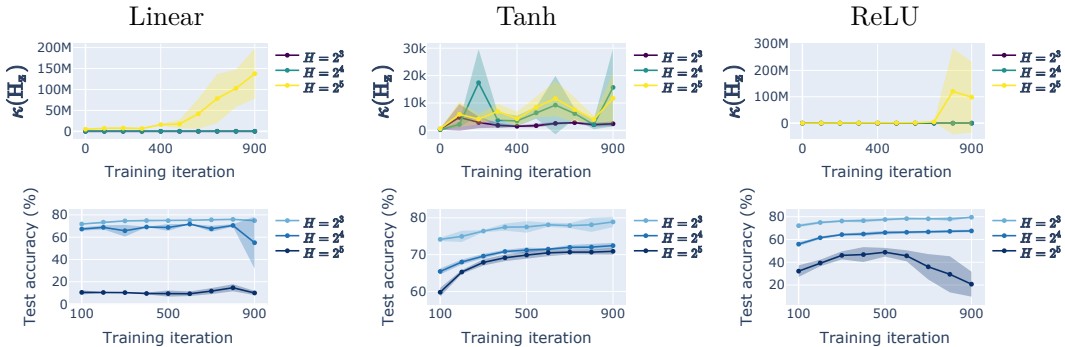

Figure A.25: **Same results as Fig. A.9 for Fashion-MNIST.**

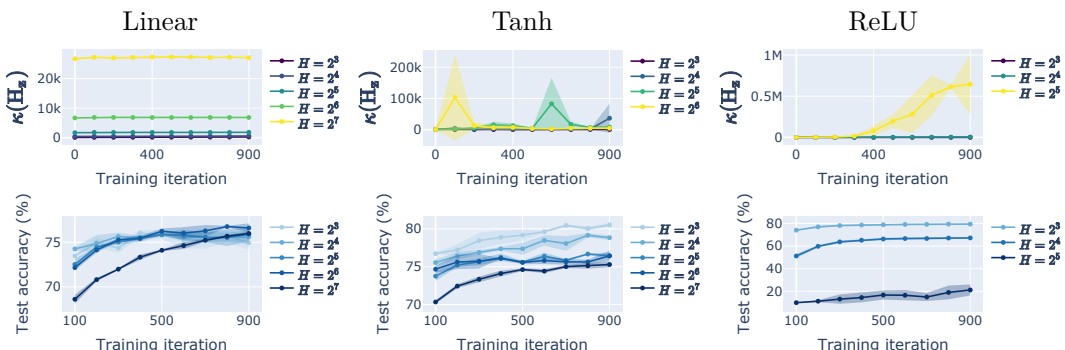

Figure A.26: **Same results as Fig. A.13 for Fashion-MNIST.**

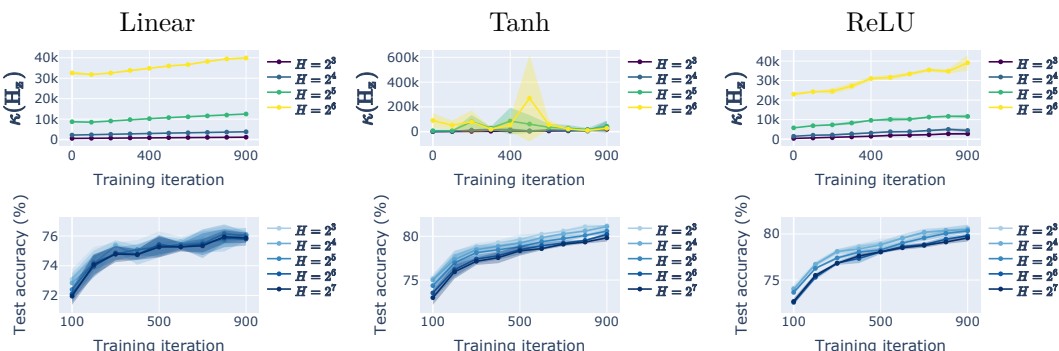

Figure A.27: **Same results as Fig. A.14 for Fashion-MNIST.**

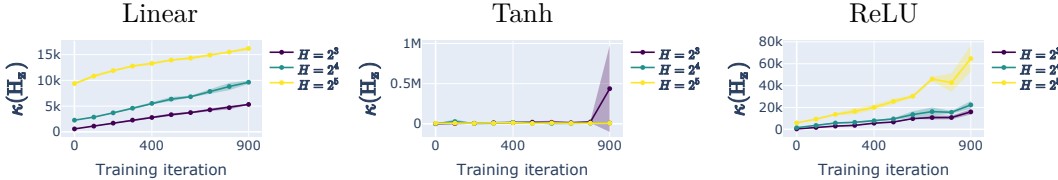

Figure A.28: **Inference conditioning during training for some $\mu$PC networks in Fig. 1.**

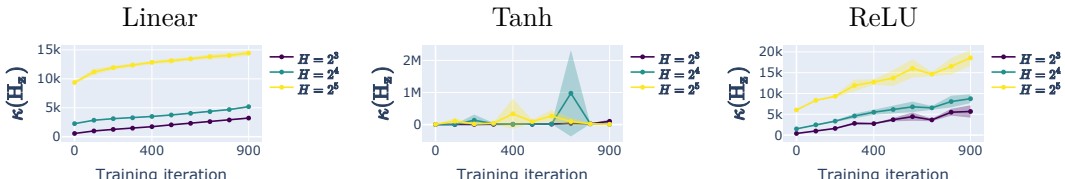

Figure A.29: **Same results as Fig. A.28 for Fashion-MNIST.**

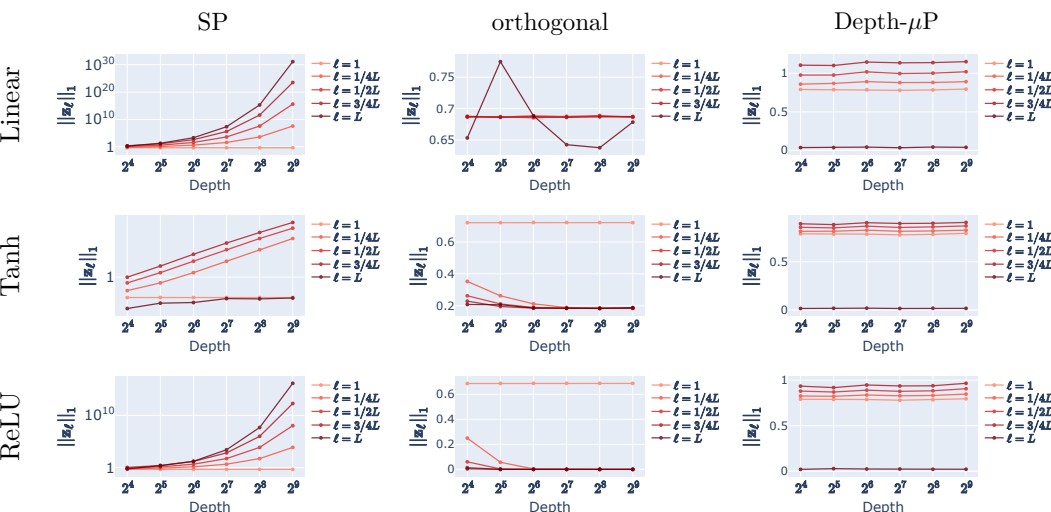

Figure A.30: **Forward pass (in)stability with network depth for different parameterisations.** For different activation functions and parameterisations, we plot the mean $\ell^1$ norm of the feedforward pass activities at initialisation as a function of the network depth $L$. Networks ($N = 1024$) had skip connections for the standard parameterisation (SP) and Depth-$\mu$P but not orthogonal. Results were similar across different seeds.

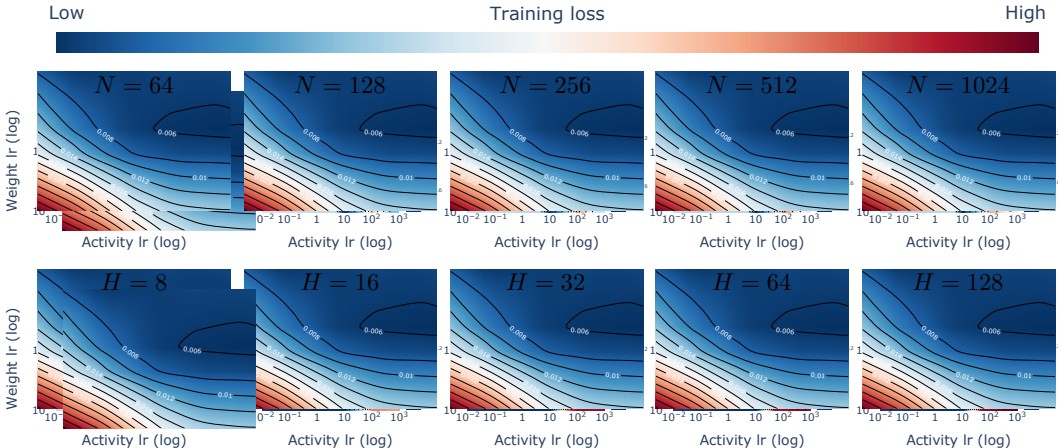

Figure A.31: **Same results as Fig. 5 for Linear.**

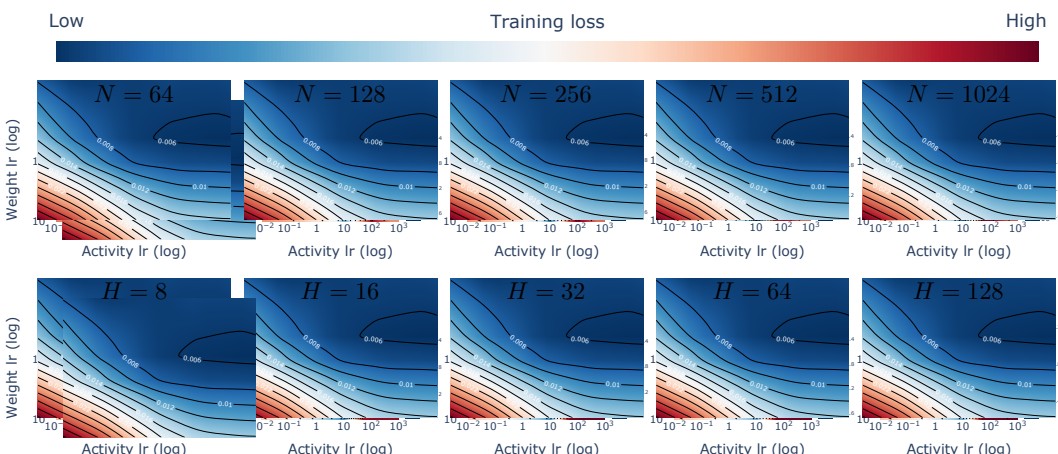

Figure A.32: **Same results as Fig. 5 for ReLU.**

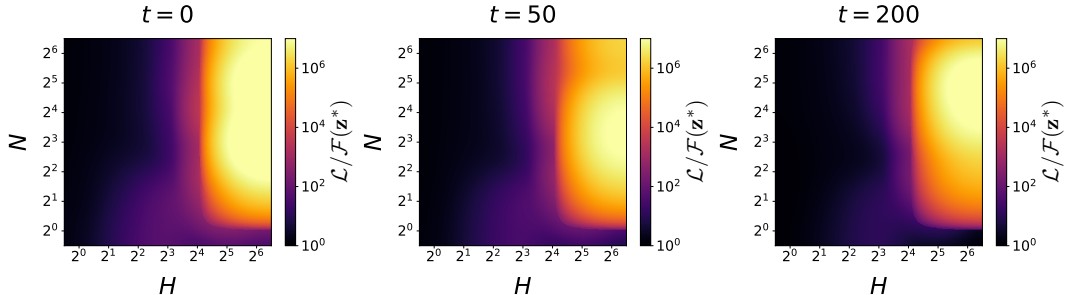

Figure A.33: **Same results as Fig. 6 for the standard parameterisation (SP).**

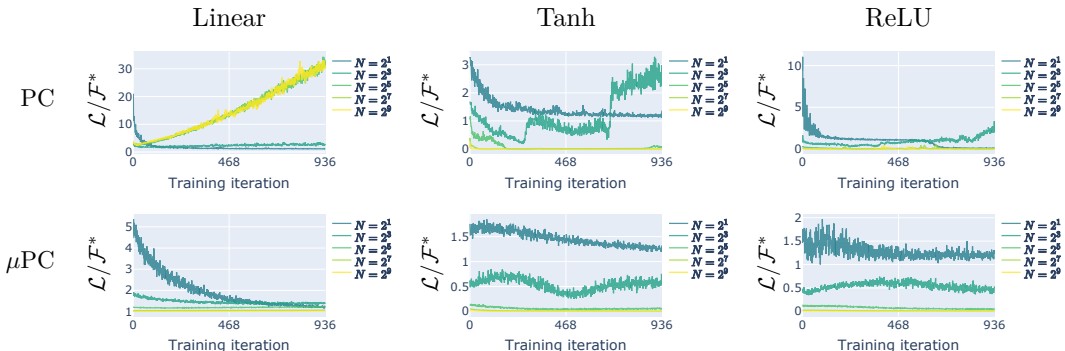

Figure A.34: **Example of the loss vs energy ratio dynamics of SP and $\mu$PC for $H = 4$.**

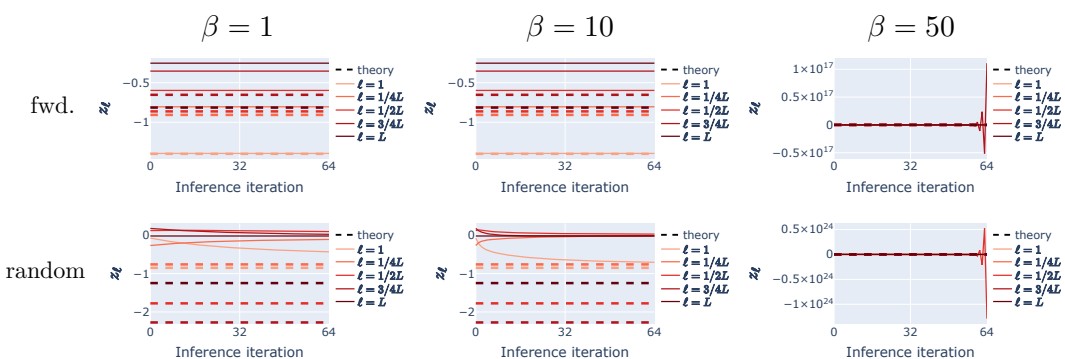

Figure A.35: **Same results as Fig. A.10 for $\mu$PC.**

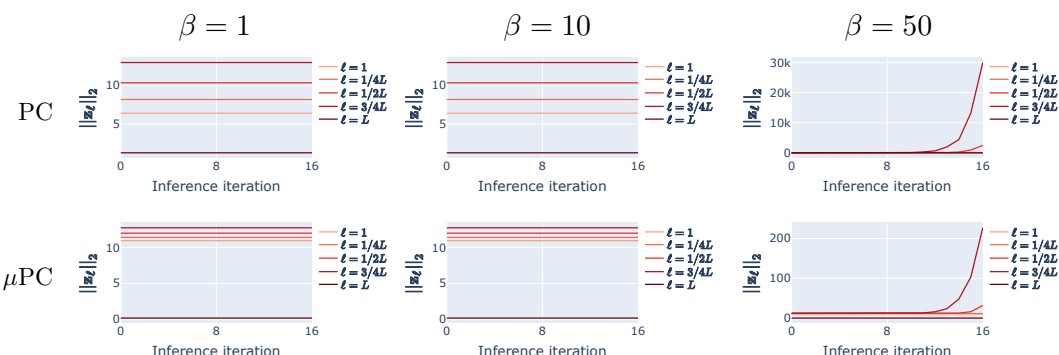

Figure A.36: **Same results as Fig. A.11 for a ReLU network.**

