# OpenReview forum: "$\mu$PC: Scaling Predictive Coding to 100+ Layer Networks"
_NeurIPS.cc/2025/Conference — NeurIPS 2025 poster_

### Official Review · Reviewer_N2W2 · 2025-06-12

**Clarity:** 3
**Significance:** 2
**Originality:** 2
**Rating:** 2
**Confidence:** 4

**Summary:**

This paper achieves stable and decent predictive coding (PC) training of deep fully connected ResNets by applying layerwise weight multipliers. These weight multipliers are motivated by Depth-muP, but not equivalent. The conditioning of the inference Hessian is evaluated as a proxy for good optimization properties, but the experiments show examples where muPC performs well despite ill-conditioned inference Hessian.

**Questions:**

- **Ill-conditioning in ResNets.** Why is the activity Hessian of ResNets so ill-conditioned? Are residual connections not expected to improve conditioning?
- **Does ReLU behave more favorably than linear or tanh?** For ReLU, the condition number does not appear to monotonically increase with depth. This is arguably the practically most relevant example. How does this behave at larger depth? This may yield important insights into the desirable properties of large ReLU nets.

**Ethical Concerns:**

["NO or VERY MINOR ethics concerns only"]

**Final Justification:**

During the discussion period, it became apparent that the parameterization used is not equivalent to depth-$\mu$P (as a function of width scaling), as suggested in the paper (e.g. Contribution 1 or Section 2.1). Since the maximal update parameterization is defined through desiderata of maximal stable updates of each trainable weight matrix, but the parameterization used does not fulfill these desiderata, the framing is misleading. I see the practical contribution of trainability of deep PC networks, but due to severe incorrectness concerns I am unable to recommend acceptance of the current form.

I find the bulletpoints in the author final remarks misleading. My correctness concern primarily critizes the wrong width dependence. muP can be realized with any width-dependent multipliers, but only in combination with the correct layerwise initialization variance and layerwise learning rate scaling. The referenced depth-muP papers may differ in their depth-dependence, but all muP-related papers agree in their width dependence. Even in Noci et al. (2024), the used parameterization only differs depth-dependently, but as a function of width a muP-equivalent parameterization is used for both SGD and Adam. The paper under review, however, uses the wrong width-dependent learning rate scaling for the chosen multipliers, and hence does not study muP, nor depth-muP, but merely some parameterization that works better than PyTorch standard. The differences of the used parameterization to muP and depth-muP in this paper are much more severe than those in the cited papers.

**Limitations:**

Some baselines are missing in the experiments. The employed scaling rule is not quite Depth-muP, thus not fully width-independent. Unlike claims in the paper, the results for $1/\sqrt{L}$ residual scaling are fundamentally limited to ResNets with shallow residual blocks. The experiments are restricted to MNIST and FashionMNIST. It remains unclear what makes the proposed parameterization stable despite diverging condition number of the activation Hessian.

**Quality:**

2

**Strengths And Weaknesses:**

**Strengths:**

To the best of my knowledge, this is the first paper to scale predictive coding (PC) to deep ResNets. The paper is written clearly and the figures are neat.

**Weaknesses:**

While I like the general idea to enable performant, deep PC training by correcting signal propagation both in forward and backward passes via depth-muP-like desiderata, the concrete realization in the paper is not ideal in several important regards. I think the paper should undergo a major revision to really unfold its full potential. I see the following important weaknesses that should be addressed:

- **Unmotivated scaling choices.** The authors claim to study Depth-muP, but then do not use the correct layerwise learning rate scaling rule if I understand lines 245-246 correctly. This choice is not motivated at all and expected to be suboptimal in terms of transfer properties. In addition, Table 1 is incomplete without specifying both the optimizer and the layerwise learning rate scaling, as SGD and Adam require different layerwise choices. As [24] already derive width-scaling muP for PC, the main novelty here would really stem from correcting depth scaling in a principled way. Is there a reason for deviating from the principled Depth-muP scaling rule?
- **Unfair comparison with SP.** It is clear that a width-independent LR induces training instability at large width. One naive learning rate scaling that could have been employed is LR= width$^{-1}$, but the practical optimal LR exponent may be larger. Arguably, a fair comparison with SP would use the optimal learning rate at each width and depth. Degeneracies at large scale are to be expected even at the optimal LR, so that Depth-muP is still expected to outperform SP under this fair and more informative comparison.
- **Usefulness of the condition number of the inference Hessian remains unclear.** The conditioning of the inference Hessian does not always seem to be a good proxy for understanding properties of the training dynamics. It remains unclear how useful it is to plot its condition number. The experiments show that muPC provides a counterexample which performs well despite bad condition number. It appears to remain open what is a good metric for predicting training instability. This could be acknowledged more clearly. Noci et al. (2024) suggests that not the Hessian but the Hessian preconditioned by the layerwise learning rate scaling is the correct metric to study, which is more directly related to the training dynamics and becomes width-independent under muP. The two metrics only coincide under the choice of multipliers under which a width-independent global learning rate can be chosen. I recommend that the authors reconsider their analysis with this metric and would expect that the conflict between Desiderata 1 and 3 is not fundamental but rather an artifact of the wrong metric.
- **Overstated generalizability.** The authors claim that their work could be extended to convolutional and transformer architectures, but this extension is not so clear. The Depth-muP paper (Yang et al., 2024) clearly states that Depth-muP is only guaranteed to work for block depth 1, so that classical convolutional ResNets and Transformers do not directly apply. In fact, recent papers show that Transformers may require the different residual scaling $L^{-1}$ (Bordelon et al., 2024; Dey et al., 2025).

**Further weaknesses in experimental or theoretical execution:**

- Figure 1 is lacking a baseline: Maybe SP also transfers across depth in this setting. If not with which exponents do its optimal (activity, weight) learning rate combination decay?
- Scaling the number of inference steps with the number of hidden layers is a potential scaling confounder. Why do you not keep this number fixed? Do the SP experiments also employ this scaling for comparability?
- Figure 6 only scales to width 64. Larger width would be interesting to understand whether larger width retains correspondence to BP up to larger depth in practice.
- In Figures 1 and 5, the color scale is lacking scalar values.
- Writing $\mathcal{O}$ in the desiderata is imprecise. Not only avoiding explosion with $\mathcal{O}(1)$, but also non-vanishing activations and their updates $\Omega(1)$ thus $\Theta(1)$ is required for muP. Otherwise, many parameterizations fulfill these weaker desiderata, which do not induce HP transfer and maximal stable feature learning in the limit.
- Theorem 1 only holds at the inference equilibrium and quickly follows from the derivation in [20]. Theory that holds throughout the optimization trajectory could hold important practical implications.

**References:**

Bordelon, Blake, Hamza Chaudhry, and Cengiz Pehlevan. "Infinite limits of multi-head transformer dynamics." NeurIPS (2024).

Dey, Nolan, et al. "Don't be lazy: CompleteP enables compute-efficient deep transformers." arXiv:2505.01618 (2025).

Noci, Lorenzo, et al. "Super consistency of neural network landscapes and learning rate transfer." NeurIPS (2024).

Yang, Greg, et al. "Tensor programs vi: Feature learning in infinite-depth neural networks." ICLR (2024).

---

> ### Author Rebuttal · Authors · 2025-07-30
>
> We thank the reviewer for their detailed feedback. Below we address their points one by one.
>
> **Scaling choice justification:** The reviewer remarks that we do not use the correct layerwise learning rate scaling rule for Depth-$\mu$P as stated in lines 245-6. The reason for this is that we could not train as deep PCNs as those tested ***without*** the scaling (of the weights and/or activities). We now make this point clear in the discussion. Nevertheless, as alluded to in the subsequent lines (246-7), it is not clear to us whether this is the version of Depth-$\mu$P that is used in practice. For example, the two original Depth-$\mu$P papers disagreed on precisely this issue, with Yang et al. (2024) but not Bordelon et al. (2023) proposing to rescale the learning rate when using Adam. We note that this inconsistency was discussed but not resolved in the review process of Yang et al. (2024). Moreover, as we note in lines 246-7, depth transfer was still achieved in two studies (e.g. Noci et al., 2024) using Adam with no learning rate scaling, on both GPT-2 and vision transformers. Regardless of the “correct” Depth-$\mu$P version, since we did not derive a novel parameterisation for PC, we simply used the Depth-$\mu$P scalings that worked best empirically. We now make this point clear in the text.
>
> Related, the reviewer remarks that Table 1 is incomplete without specifying both the optimiser and the learning rate. This is correct, and these details were omitted because we used Adam for all the experiments with no learning rate scaling, as explained above and in lines 245-6. Again, the reason was that this is the only parameterisation that we found to lead to stable training of very deep PCNs. We now make this clear in the main text.
>
> **Comparison with SP:** The reviewer notes that we did not use a width-dependent scaling for the learning rate of SP, making the comparison with $\mu$PC unfair. We make two points in response. First, we performed an extensive grid search (line 758) that in practice covered many width exponents given that the networks in Figure 5 had width $N = 512$. Second, and more to the point, the transfer results were not the focus of our work given that we do not derive a novel parameterisation for PC, and there was therefore no reason to expect transfer. This is a rather surprising result that needs to be better understood. We agree that any parameterisation specific to PC should perform a more rigorous analysis.
>
> **Usefulness of the condition number of the inference Hessian:** The reviewer questions the usefulness of this metric based on the results that networks can be trained with PC or $\mu$PC despite high ill-conditioning of the inference landscape. We agree that these results are puzzling but argue that the inference conditioning has still an important impact, even if in a subtle way. In particular, taken together, many results of the paper show that (close) convergence of the inference dynamics to a solution is ***necessary for trainability*** and, therefore, that understanding of the landscape conditioning is crucial. The clearest demonstration is an experiment where we trained standard deep linear PCNs with “perfect inference” (using the solution of Eq. 4). As shown in Figure A.15, we find that, even without $\mu$PC, very deep PCNs become trainable in the sense that the training loss decreases monotonically. These results suggest that ***if we could reach the solution*** (which is practically impossible at large depth), the networks would be trainable. (To prevent further misunderstandings, please note that trainability is necessary but not sufficient for good generalisation, as shown in Figure A.15.)
>
> Why does $\mu$PC work despite the ill-conditioning? It turns out that ***$\mu$PC initialises the activities much closer to the solution than standard PC*** (see experiment in Figure A.11). At the same time, we find that $\mu$PC still needs as many inference steps as hidden layers to obtain good performance (see Figure A.14 for results with one step), showing that it does not initialise exactly at the solution. Together, these results show that (i) close inference convergence is necessary for stable training, that (ii) this becomes hard at large depth because of the ill-conditioning, and that (iii) having a stable forward pass is a way of initialising much closer to the solution. Without an understanding of the inference landscape, these results would be much harder to reconcile. For space reasons, this discussion was not included in the paper but will be added to the camera-ready version.
>
> Having clarified the importance of the conditioning of the inference Hessian, we agree that an analysis of the weight Hessian as performed by Noci et al. (2024) could be useful and complementary to our study. However, we would like to highlight that it is impossible to disentangle the weight dynamics from the inference dynamics, so any analysis of the former would have to make assumptions about the latter. Indeed, this what was done in Innocenti et al. (2024), who studied the weight Hessian of PC ***at the inference equilibrium***, showing that it has some better properties compared to the mean squared error loss of feedforward networks. However, these properties broke down empirically on deep PCNs, showing that an equilibrium was no longer reached in practice. For all these reasons, we do not think that the conflict between Desiderata 1 and 3 is an artifact but could be an inherent limitation of such local models.
>
> **Overstated generalizability:** The reviewer points out that our claim about extending the work to convolutional and transformer architectures is overstated. Here we were being loose with the phrase “could be extended” and simply meant that Depth-$\mu$P as developed for CNNs and transformers could be applied to PC in the same way we did for fully connected ResNets. We agree with the reviewer that our original wording can be misinterpreted to overstate our results, so we qualify this and similar statements in the paper to mean that our work can be seen as a first step towards a more stable parameterisation of deep PCNs.
>
> **Further weaknesses**
>
> To address further weaknesses pointed out by the reviewer:
> * We do not have a transfer plot for SP in Figure 1 because as noted above standard PCNs become untrainable at such depths.
> * The reason why we scale the number of inference steps with the number of hidden layers is that one needs at least $H$ steps for the inference (and therefore weight) gradients to propagate (i.e. become non-zero) to all the layers. The easiest way to see this is to note that at the forward pass all the prediction errors except the last one are zero, since the PC energy collapses to the mean squared error loss. The experiments with SP also employ this scaling of the inference steps, and here it is also worth noting that in practice more inference steps are often used, which is consistent with the point made above that the forward pass of standard PC initialises the activities far from a solution.
> * Networks with $H = N = 64$ were the widest and deepest we could compute the theoretical energy for, due to a naive (i.e. inefficient) computation of the activity Hessian. However, we will add plots to the appendix for small depth and larger width similar to Figure A.31.
> * We omitted scalar values from the transfer results of Figure 1 and 5 for a less cluttered visualisation. They do not materially affect the results.
> * We agree that using $\mathcal{O}(1)$ is imprecise and now correct the text to mean $\Theta(1)$.
> * We agree that theory that holds throughout the training dynamics of PC would be important and potentially very impactful. As noted above, this is challenging since the PC learning dynamics are hard to characterise far from the inference equilibrium. Moreover, even at the equilibrium, the analytical expression for the equilibrated energy is non-trivial (Eqs. 30-31). Nevertheless, this is clearly an important direction for future work.
>
> **Questions**
> * The reviewer asks why the activity Hessian of ResNets is so ill-conditioned. This is a good question, especially since it is known that ResNets help with the ill-conditioning of the weight landscape as noted by the reviewer. We have no rigorous explanation for why ResNets lead to more ill-conditioned inference, mainly because the activity Hessian is challenging to study even at initialisation from a random matrix theory perspective (Section A.2.3). Intuitively, however, if one examines the structure of the ResNet Hessian (Eq. 24), one realises that the off-diagonal terms dominate the diagonal ones much more so than without skip connections (Eq. 5). A slightly different way of saying this is that ***ResNets increase the network interactions*** (by adding more connections), and as noted in lines 183-5 a naive way to make the Hessian perfectly conditioned is to remove all interactions.
> * It is not entirely clear to us why the reviewer remarks that for ReLU networks the condition number does not appear to monotonically increase with depth. The only plot suggesting this is Figure 3 where at depth 32 the condition number increases more slowly than at half the depth. We note that the conditioning is >10k compared to the Linear and Tanh plots (>1k), and all other relevant plots on ReLU networks show that the conditioning still grows with the depth ***at initialisation*** with or without skip connections (Figs. 2, 4 and A.21).
>
> **References**
>
> Yang, Greg, et al. "Tensor programs vi: Feature learning in infinite-depth neural networks." ICLR (2024).
>
> Bordelon, Blake, et al. "Depthwise hyperparameter transfer in residual networks: Dynamics and scaling limit." arXiv preprint arXiv:2309.16620 (2023).
>
> Noci, Lorenzo, et al. "Super consistency of neural network landscapes and learning rate transfer." NeurIPS (2024).
>
> Innocenti, Francesco, et al. "Only strict saddles in the energy landscape of predictive coding networks?" NeurIPS (2024).

---

> > ### Comment · Reviewer_N2W2 · 2025-08-06
> >
> > I thank the authors for the detailed response. I have some clarification questions:
> >
> > 1. I understand that the learning rate is not rescaled as a function of depth. Is it correct that the weight multipliers specified in Table 1 were used, and the learning rate was held constant across widths for both training the parameters with Adam and the inference with GD?
> > 2. Could the authors kindly clarify what they mean with the statement ‘The reason for this is that we could not train as deep PCNs as those tested ***without*** the scaling (of the weights and/or activities).’ in the context of lacking layerwise learning rate scaling?

---

> ### Author Response · Authors · 2025-08-06
>
> To address the reviewer's questions:
> * It is correct that we used the weight multipliers specified in Table 1 and that the learning rate was held constant across widths for both the training of the parameters using Adam and the inference steps of the activities using GD. In brief, we do not rescale any learning rate in any way.
> * By that sentence, we meant that if we tried to rescale the step size/learning rate of *either* or *both* the parameters (updated with Adam) or the activities (updated for inference with GD) by $1/ \sqrt{NL}$—as recommended by (at least some version of) Depth-$\mu$P—the networks performed at near-chance accuracy and showed no improvement with training.
>
> We hope this clarifies our parameterisation and findings.

---

> > ### Comment · Reviewer_N2W2 · 2025-08-07
> >
> > Thank you for the clarification. My concerns about the comparison with SP, the overstated generalizability, miss-formulated desiderata and my questions have been addressed.
> >
> > I find the insights into the ResNet Hessian off-diagonal terms interesting. My point about the raw inference Hessian condition number questioned that this is the ideal metric for capturing ‘convergence of the inference dynamics’, and your experiments showing good performance despite large condition number support this statement (as acknowledged in e.g. l. 688-691). I am not entirely convinced by your arguments concerning this point, but I see that a detailed analysis of the training dynamics away from the inference equilibrium would be challenging.
> >
> > Replacing ‘low’ and ‘high’ by scalar values in Figures 1 and 5 would provide relevant information. For example, is the training set ever interpolated, or how far is the range of training errors here? I encourage including scalars in the revision.
> >
> > Most importantly, I believe the employed scaling is not depth-muP equivalent neither with respect to width nor depth scaling. For the correct width scaling, with the choice of weight multipliers provided in Table 1, the SGD learning rate would need to be rescaled by $\cdot width$ (see e.g. Tensor Programs V Table 3 and the equivalence relation in Lemma J.1, but also Table 1 in Bordelon et al. ‘23), and the Adam hidden-layer learning rate would need a factor $\cdot width^{-1/2}$ whereas input and output layers can have width-independent learning rate from analogous arguments.
> >
> > As a consequence, I believe the employed parameterization is not muP-equivalent, does not induce ideal width-independence nor depth-independence, empirical transfer may break at sufficient but realistic widths, but would improve with the correct scaling rule. As a $\mu$P-like parameterization ia a crucial claimed component of this paper, hence I cannot recommend acceptance and will keep my score.

---

> ### Author Response · Authors · 2025-08-07
>
> We thank the reviewer for their comments and are glad that we addressed some of their concerns.
>
> Regarding the utility of the conditioning of the inference Hessian, we agree that it does not capture convergence of the inference dynamics per se, but more precisely their *convergence speed*. This is why, as explained in our previous response, $\mu$PC seems to work despite large condition number, namely, it initialises the activities much closer to a solution and so ill-conditioning becomes "less of a problem". This is also supported by the experiments with ResNets without $\mu$PC which, while still being trainable, show much worse performance than $\mu$PC. These results clearly show that, *if the landscape were well-conditioned*, deep PC networks would be trainable regardless of their forward pass stability. Therefore, we think that this is a valid desideratum to have. It is also an important finding for the future study of non-standard architectures that might induce better conditioning.
>
> We agree that scalar values for Figures 1 and 5 would be informative and so we will update the figures accordingly.
>
> Regarding the Depth-$\mu$P scalings, we agree with the reviewer that our parameterisation is strictly not equivalent to Depth-$\mu$P *with respect to the width scalings* for the reasons they mention, while we respectfully disagree that it is not equivalent to Depth-$\mu$P *with respect to the depth scalings*.  This is because, as explained in our previous response, the two original Depth-$\mu$P papers disagreed on the learning rate scaling of Adam, and depth transfer has been shown without this specific scaling in 2 different studies using complicated architectures.
>
> Nevertheless, we think that ***the focus on the equivalence of our proposed parameterisation to Depth-muP is misplaced and fundamentally misunderstands the contributions of our work***. Because PC networks have non-trivially different learning dynamics than backprop-trained networks (at or far from inference equilibrium), it is incorrect to claim that $\mu$PC induces "ideal width-independence nor depth-independence" and that empirical transfer "would improve with the correct scaling rule". This is because the Depth-muP scalings were derived for backpropagation (BP), not for PC, and only the results about the forward pass at initialisation transfer from one algorithm to the other. Therefore, ***the "correctness" or optimality of the original Depth-$\mu$P scalings is no longer guaranteed during training of PC networks***. Indeed, we think it is quite telling that it is precisely the learning rate scalings that seem to differ in optimality between PC and BP.
>
> The main contribution of this work ultimately lies in enabling the training of deep PC networks, which was not possible until now and had been recently posed as an important challenge for the community by Pinchetti et al. (2024). Indeed, as we note in the text, to our knowledge 100+ layer networks had *never* been successfully trained before with *any local learning algorithm* even on simple datasets. The fact that this was achieved without the all the scalings of Depth-$\mu$P - *originally derived for BP* - is in and of itself interesting and motivates the study of parameterisations specific to the learning dynamics of PC. It is equally interesting that empirical transfer was achieved *without any theoretical guarantees*, which also warrants further study. For these reasons, we hope that the reviewer will reconsider their decision.

---

> > ### Comment · Reviewer_N2W2 · 2025-08-08
> >
> > I thank the authors for the additional clarifications. I see that deriving the actual ‘maximal update parameterization’ for PC would require a more detailed analysis to derive all maximal stable learning rate scalings and weight multipliers. But this questions using the term $\mu$P so prominently in the paper even more. Neither $\mu$P for the traditional setting of backprop is used (even though the writing suggests that depth-$\mu$P is used), nor is a parameterization derived for PC that fulfills the ‘maximal update’ desiderata that originally define $\mu$P.
> >
> > I acknowledge the practical achievements you list, in particular scaling PC trainability to deep networks, but when framing the paper prominently in the light of $\mu$P, I would expect a parameterization that fulfills the $\mu$P desiderata. Therefore I respectfully disagree with missunderstanding the contributions and cannot recommend acceptance.

---

### Official Review · Reviewer_qKFu · 2025-06-24

**Clarity:** 2
**Significance:** 2
**Originality:** 3
**Rating:** 4
**Confidence:** 4

**Summary:**

The article addresses a major issue in biological network training: it is well-known that almost all of the biologically plausible approaches suffer from depth-scalability problem. This article proposes the adoption of the Depth-$\mu P$ scheme, which initializes the network weights based on the width and depth of the network. The Depth-$\mu P$ scheme was proposed mainly for hyperparameter transfer capability for backpropagation training of networks. The current article wisely adopts this approach to overcome the difficulties of training predictive coding (PC) based biologically plausible networks, with the additional gain of enabling hyperparameter transfer for PC training, as originally devised for BP training. The article starts with a summary of the relevant work and the PC loss description. Then it provides an analysis of the unstability of PC based training based on the analysis of the loss function and the corresponding algorithm. Later, the use of Depth-$\mu P$ parametrization is proposed to enable stable training and hyperparameter transfers. The article provides numerical experiment results for the MNIST dataset, to demonstrate the successful training of deep networks using PC approach, and the hyperparameter transfer capability.

**Questions:**

-  Could you clarify why you use a different label/acronym ($\mu PC$) for Depth-$\mu P$? The footnote in Page-3 references  Section 4 to clarify this distinction, but when you check Section 4 there is no satisfactory explanation. It is clear that PC loss and the MSE loss of BP are not the same but in terms of the initialization/parametrization ($\mu PC$) is exactly equal to Depth-$\mu P$. This, in my opinion, creates confusion about whether there is any change specific to the adoption of Depth-$\mu P$ to the predictive coding.

- Is this approach extendable to more complex classification tasks?

**Ethical Concerns:**

["NO or VERY MINOR ethics concerns only"]

**Final Justification:**

I believe  the article presents a valuable contribution to  the problem of scalability of biologically plausible networks, by using results from theoretical scaling analysis of networks. However, I am a bit concerned about the restriction of numerical examples to MNIST. During the discussion period, I stated this concern, hoping that the authors could share some preliminary results on a different data set such as CIFAR-10. However, the authors responded that they will share results in the camera ready paper. This is a bit concerning, I believe it should not be relatively hard to generate results for CIFAR-10 dataset during the discussion period. I increased my score to 4, but I still feel not comfortable about the lack of further experimental support for the proposed valuable idea.

**Limitations:**

Yes.

**Paper Formatting Concerns:**

I listed some presentation-based issues above. Otherwise, the article seems to be compliant with formatting instructions.

**Quality:**

3

**Strengths And Weaknesses:**

## Strengths

- It is demonstrated that the proposed parametrization approach can be used to successfully train $128$-depth networks based on the predictive coding loss for the MNIST datset. Scaling biologically plausible neural networks is an important research area, therefore, this result raises the hopes by underlining the importance of the initialization schemes in stable training of deep biological networks.
- The hyperparameter transfer is the main advantage of the existing Depth Depth-$\mu P$ based parametrization. In this work, it is demonstrated to be applicable to the predictive coding based neural network training.

## Weaknesses
- One of the main drawbacks of the article is to assess whether the depth scaling property demonstrated for the MNIST dataset generalizes to other data sets. Even CIFAR-10, CIFAR-100 would be informative for this purpose.
- The article's presentation requires significant improvement. In particular, the  positioning of the article's contributions relative to the existing work and its novelty are  obscured by the terminology and the presentation style of the article. For example, the use of the $\mu PC$ acronym to replace the existing Depth Depth-$\mu P$ causes confusion and  understanding the actual contributions. Many of the figures in appendix are lacking captions and discussions/explanations. In this sense, the article requires a major rewrite which would clarify the main contribution points

## Additional Comments
- it is better fan_in is defined for clarity.
-  Table 1 could be placed in the second page, where it is first referenced.
-  Using an extra symbol (H) for $L-1$ is unnecessarily confusing.
- zero-shot transfer figures in Figure 1 (left-down corner) are not clearly explained in the Figure caption. I was not able to see referrals to this figure part in the article's discussion.

---

> ### Author Rebuttal · Authors · 2025-07-28
>
> We thank the reviewer for their feedback and fair summary of our work. Below we address their points one by one.
>
> **Limited experiments**
>
> First, we completely agree that the main weakness of our work are the limited experiments testing $\mu$PC, specifically the range of datasets used. The main reason for this is that training of such deep models is highly computationally expensive on GPUs even on simple datasets such as MNIST, due to both the size of the networks and the sequential nature of PC inference. We would also like to point out that, as discussed in the Experiments section, we also tested $\mu$PC on Fashion-MNIST classification. Nevertheless, we agree that it would be important for future work to test $\mu$PC on more complex datasets and architectures, as mentioned in the conclusion.
>
> To directly answer one related question asked by the reviewer, it seems reasonable to expect that $\mu$PC should extend to more complex classification tasks given that convolutional neural networks (CNNs) parameterised with Depth-$\mu$P simply involve a different width scaling. Nevertheless, this needs to be empirically shown, and there are other subtleties related to CNNs highlighted by some previous Depth-$\mu$P work (Bordelon et al., 2023)—such as how to scale hidden layers outside the residual blocks—that might require further changes to the parameterisation for PC. This is why we think that, as noted in the conclusion, trying in the first instance to extend $\mu$PC to CNNs is one of the most exciting future research directions.
>
> **Clarification of related work:** The reviewer points out that the paper’s summary of contributions could be better positioned relative to previous work. As noted in the text (line 44), all related work and its relationship with our contributions is reviewed in detail in the appendix (from line 445) due to limited space. The reason why our summary of contributions does not directly mention or relate to any previous work is that all the results presented are novel and go significantly beyond the few existing works on training deep PCNs (reviewed in the related work section).
>
> **Clarification of the $\mu$PC acronym:** The reviewer notes that the $\mu$PC acronym causes confusion and misunderstanding of the contributions. This was not at all our intention. We used the acronym $\mu$PC to encapsulate ***both the algorithm and the parameterisation***. As noted in footnote 3 and as the reviewer correctly points out, this was in part to distinguish the different objectives of PC (energy) and BP (MSE loss) even though the parameterisations are the same. However, it was also to avoid repeating ***PC with Depth-$\mu$P***, which would have been often necessary to distinguish it from ***BP with Depth-$\mu$P***. We hope this clarifies the acronym’s usage and now make it more clear in the relevant footnote. In terms of the contributions, we would like to stress that $\mu$PC is essentially “just” (one version of) Depth-$\mu$P applied to PC and that it was not directly developed from analysing PC. We think that our wording throughout the paper reflects this, where we never claim that we propose a novel parameterisation and often state that $\mu$PC reparameterises PCNs using Depth-$\mu$P.
>
> **Clarification of missing captions and explanations:** The reviewer points out that many of the figures in the appendix are missing captions and explanation. These omissions are deliberate and, we believe, appropriate. For example, we discuss all the figures in the Additional experiments section (A.3) and omit captions whenever the experimental details are the same as those of another figure with a detailed caption (and this figure is always referenced in the title). This is true (for example) for many of the figures plotting the condition number of the activity Hessian over training (Figs. A.8-9 and A.22-26), which share the same setup as Figure 3 in the main text of the paper and only change one variable (e.g. dataset or architecture) that is always clearly stated in the figure title. Many of the Supplementary figures in the appendix (A.6) share this feature.
>
> **Additional comments**
>
> To address the reviewer’s additional comments:
> * We agree that “fan_in” was confusing and now replace it everywhere with $N_\ell$ for appropriate $\ell$.
> * In the writing of this paper, we received mixed feedback as to the positioning of Table 1, with some suggesting (like the reviewer) to place it where it is first referenced (Section 2) while others recommending the Desiderata section. We think the latter is the most appropriate because this is the place where we motivate and introduce $\mu$PC.
> * We agree that the variables $L$ for number of layers or weight matrices and $H = L-1$ for number of hidden layers can be confusing, especially since we use them both to refer to the network depth (as noted in lines 99-100). We found this impossible to avoid without being mathematically imprecise given that (i) $L$ features in the Depth-$\mu$P scalings, and (ii) many of the quantities related to the PC activities involve $H$ (as there are $H$ hidden layers of states free to vary).
> * We agree that the zero-shot transfer results in Figure 1 are not explained in the caption; however, we immediately after (last two lines of the caption) refer to the full transfer results for explanation. We changed the wording to make this more clear.
>
> **Summary**
>
> To summarise, we completely agree with the reviewer that the limited experiments testing $\mu$PC is the main weakness of our work and that this is an important (and exciting) future direction. In this context, we would like to highlight the extensive range of other experiments performed to arrive at the conclusion that $\mu$PC could help with training at large depth, especially on the conditioning of the inference landscape and other parameterisations (i.e. orthogonal initialisation). We also hope that we clarified the use of the $\mu$PC acronym and our contributions, as concerns raised by the reviewer. The strength of our work ultimately lies in clearly identifying problems with the standard parameterisation of deep PCNs and exploring what we call $\mu$PC as a first step towards a more stable and scalable parameterisation. We strongly believe that the local learning community would likely benefit from our findings.
>
> **References**
>
> Bordelon, Blake, et al. "Depthwise hyperparameter transfer in residual networks: Dynamics and scaling limit." arXiv preprint arXiv:2309.16620 (2023).

---

> > ### Comment · Reviewer_qKFu · 2025-08-02
> > **Thanks for your response**
> >
> > I would like to thank the authors for their response. Overall, I believe that adapting theoretically derived initialization schemes to biologically plausible network training for scalability is a valuable contribution. I also found the authors’ clarifications and proposed improvements to the presentation generally satisfactory.
> >
> > However, I believe the paper would benefit a lot from at least some preliminary results on the CIFAR-10 dataset, which has a similar input dimensionality to MNIST but presents a more challenging classification task. Including such results would provide stronger support for the generalizability of the experimental findings on network depth scalability.

---

> > > ### Author Response · Authors · 2025-08-02
> > >
> > > We thank the reviewer for their comments. We agree that some preliminary results on CIFAR-10 would be insightful to better assess the generality of the findings so we will work to include these in the camera-ready version of the paper.

---

### Official Review · Reviewer_ijZ5 · 2025-06-24

**Clarity:** 2
**Significance:** 3
**Originality:** 3
**Rating:** 5
**Confidence:** 5

**Summary:**

The paper *"µPC: Scaling Predictive Coding to 100+ Layer Networks"* presents µPC, a novel parameterisation that enables scalable training of deep predictive coding networks (PCNs) with up to 128 residual layers.
It identifies that traditional PCNs face challenges with depth in residual architectures due to ill-conditioned inference initialisations. µPC, leveraging Depth-µP scaling, addresses these issues, enabling stable training, competitive performance, and zero-shot transfer of learning rates across varying network widths and depths.
Beyond these results, the authors provide extensive analysis of four desiderata for stable parameterisations, showing that satisfying the first (desideratum 1) alone suffices for stable training in residual networks.
The additional desiderata and corresponding analyses offer future research directions and tools for enhancing training stability in PCNs.
Overall, this work marks a step toward scaling local, biologically plausible learning algorithms to the level of modern deep learning systems.

**Questions:**

Major

- Is the proposed method only applicable to residual networks? Figures A.17 and A.18 suggest that deep ReLU networks benefit from µPC. What exactly are these ReLU networks? Do they include residual connections? If not, which initialisation do they use—is it the µP initialisation described in Section 2.1 or the Depth‑µP initialisation proposed specifically for ResNets? Clarifying this will make the scope of the work clearer.

Minor

- In Section 2.2, you state that SP uses a normal initialisation with variance 1/fan_in. However, in your implementation, it appears that you use a uniform initialisation with the same variance, as you rely on equinox.nn.Linear (https://github.com/patrick-kidger/equinox/blob/main/equinox/nn/_linear.py) which uses a uniform initialisation from (https://github.com/patrick-kidger/equinox/blob/main/equinox/nn/_misc.py). Did I misinterpret your codebase? If not, does the choice between uniform or Gaussian initialisation materially affect your results or analysis?
- You devote a lot of space to the ill-conditioned energy landscape, arguing that this poses a challenge for PC inference. However, from Figures 2 and 4, it appears that µPC results in a significantly more ill-conditioned landscape than standard PC, with κ(Hz) increasing by factors of 1,000 to 10,000. Do you think the improved training stability observed with µPC comes at the cost of slower inference, particularly if you let inference run until convergence instead of fixing the number of inference steps to the depth of the model?

**Ethical Concerns:**

["NO or VERY MINOR ethics concerns only"]

**Final Justification:**

I have read the authors' rebuttal, which has addressed my earlier questions. Although I still have reservations regarding whether this method will lead to improved learning performance on more challenging tasks, I am now convinced that this work makes a significant contribution to the predictive coding literature. Consequently, I have increased my confidence score.

**Limitations:**

yes

**Paper Formatting Concerns:**

I have no paper formatting concerns.

**Quality:**

3

**Strengths And Weaknesses:**

## Strengths

Quality

- Extensive experimental evaluation: The authors demonstrate stable training of predictive coding networks with up to 128 residual layers on classification benchmarks, supporting their claims with empirical evidence.
- Clear scaling analysis: They conduct extensive scaling studies, identify two pathologies in standard PCNs, and show how µPC addresses one of them through Depth‑µP parameterisation.

Clarity

- Contributions were clearly stated
- Figures were clear.

Significance

- Unlocking depth stability for residual PCNs, potentially enabling biologically plausible learning in very deep models.
- Zero‑shot learning‑rate transfer: Demonstrating that learning rates generalise across network widths and depths is practically valuable for reducing hyperparameter tuning in future PCN research.
- The proposed techniques likely extend to residual networks with convolutions and other base operations, broadening the paper's impact.

Originality

- Applying Depth‑µP scaling to predictive coding is a novel approach that bridges recent developments in parameter scaling with biologically plausible learning. While µP has appeared in other contexts, its integration with predictive coding for depth scaling is innovative and opens new research directions for predictive coding.

## Weakness

Quality

- Evaluated on simple classification tasks that do not require deep architectures. Training on MNIST and Fashion MNIST typically does not require more than 2 hidden layers. While the parametrisation improves stability, it remains unclear whether any layers beyond the final few actually learn. Thus, the applicability of µPC to more complex tasks that require depth remains uncertain.
- Partial stability fix: The proposed initialisation ensures stability only at the start of training. It remains unclear whether the model remains stable throughout learning, especially for more complex architectures.

Clarity

- The proposed method µPC is difficult to locate in the main text on first read and requires a deep dive to understand, despite being a relatively simple change. A clearer structure with a dedicated section for the desiderata and another for the proposed parametrisation would improve accessibility.
- Figure A27 should be moved into the main text to illustrate the problem µPC aims to solve.
- Figure captions should be improved to make figures stand-alone—for instance, in Figure 2, the activity Hessian κ(Hz) should be explained (e.g., lower values are better).
- The paper is generally difficult to read and navigate. While it condenses extensive work into 9 pages, it would help to move the ill-conditioned landscape results (Figures 2–4) to the appendix or combine Figures 2 and 4. These could then be replaced with results that directly reflect the properties µPC targets (e.g., Figure A27), helping reader understand what muPC is.

Significance

- While the work is significant for predictive coding research, backpropagation still outperforms PC on the evaluated tasks. Exploring learning tasks where PC has advantages over backpropagation could yield more insightful comparisons.

Originality

- The core change to PC parametrisation is directly drawn from the existing µP literature. Adapting this parametrisation specifically to PC inference dynamics and weight updates could lead to more substantial advancements for predictive coding.

---

> ### Author Rebuttal · Authors · 2025-07-29
>
> We thank the reviewer for their detailed feedback and what we think is a very fair assessment of the work. Below we address the points made by the reviewer one by one.
>
> **Quality**
>
> * We completely agree that the tasks we tested $\mu$PC on were simple and do not require deep architectures. We further agree that, as noted in the conclusion, it will be important (and exciting) to try to extend, in the first instance, $\mu$PC to convolutional networks so that performance on more complex datasets requiring depth can be properly assessed.
> * The reviewer points out that $\mu$PC ensures stability only at the start of training. We see two possible interpretations of this statement. The first is that $\mu$PC does not explicitly satisfy Desideratum 2 about the stability of the forward pass during training. It is true that there is no theoretical guarantee provided; however, the empirical results clearly show that $\mu$PC maintains good performance throughout training, suggesting (although not proving) that Desideratum 2 is satisfied. This point was not in the paper for space reasons but will be added to the camera-ready version. The second interpretation of the reviewer's point has to do with results of Section 6 (Theorem 1 and Figure 6) where the correspondence between $\mu$PC and BP breaks down during training. Here we would like to highlight that if an algorithm does not approximate the BP gradients, it does not necessarily mean that the algorithm cannot lead to stable training dynamics or good performance. For PC, this was clearly demonstrated by Innocenti et al. (2024), who showed that PC in fact performs a completely different gradient update at the inference equilibrium and that, under certain conditions, this can even lead to faster convergence. We revised Section 6 to make this point clear and avoid potential confusion. Having said all this, we would like to emphasise that we do not claim that $\mu$PC is completely stable. $\mu$PC clearly does not fix the ill-conditioning of the landscape (at initialisation or during training), and in this sense the parameterisation remains unstable.
>
> **Clarity**
>
> To address the reviewer’s points about the clarity of the paper:
> * We appreciate that the paper condenses a lot of information and that $\mu$PC can be difficult to locate. The camera-ready version of the paper will aim to clarify the presentation of $\mu$PC.
> * We did not include Figure A.27 in the main text because it is not a novel result but, as noted in line 172, a ***replication*** of the stability of the forward pass with depth ensured by the Depth-$\mu$P and orthogonal parameterisations. Despite the importance of these results to our work, we felt that including this figure in the main text could have led to a misunderstanding of our contributions.
> * We clarified the captions of the figures in the main text to make them more stand-alone.
>
> **Significance**
>
> The reviewer remarks that backpropagation still outperforms PC on the evaluated tasks. Could they please clarify what is meant by this? Our empirical results on $\mu$PC clearly show that performance is not significantly different from BP, both during training and at convergence, for both MNIST and Fashion-MNIST (see Figs. 1 & A.16-18). We suspect that this could be due to a misunderstanding of Figures A.17-18, which the reviewer also asks about. To answer the specific questions raised:
> * $\mu$PC does indeed only work for residual networks (ResNets) because ***Depth-$\mu$P was in turn developed for ResNets***. We mention this in the main text (lines 82 and 172-3), and a brief explanation is included in the related work section on $\mu$P (from line 486). Essentially, this is because the forward pass of standard MLPs is hard to stabilise with respect to the depth without skip or residual connections.
> * Figures A.17-18 are ***residual*** ReLU networks trained with $\mu$PC. We thank the reviewer for pointing out this imprecision and now clarify in the figure titles that these networks have skip connections. As discussed in the Experiments section (lines 210-11), Figure A.17 shows the performance of the same ReLU networks of Figure 1 trained on MNIST until convergence (5 epochs). This is also noted in the caption of Figure 1. Figure A.18 shows the same results as Figure A.17 on Fashion-MNIST. We hope that this helps clarify the significance of our results.
>
> **Originality**
>
> We completely agree with the reviewer that developing a parameterisation that is adapted to the specific inference and learning updates of PC would likely lead to further improvements. Here we would like to note that this is challenging in part because the PC learning dynamics are hard to characterise far from the inference equilibrium. Moreover, even at the equilibrium, the analytical expression for the equilibrated energy is non-trivial (Eqs. 30-31). Nevertheless, we think that theoretical work would be important and impactful here.
>
> **Questions**
>
> To address the reviewer’s minor questions:
> * We indeed use the Kaiming/He uniform initialisation for all of the experiments except for those on the random matrix theory of the activity Hessian (Section A.2.3), where (for theoretical purposes) we use a Gaussian with variance scaled by 1/fan_in. This initialisation for SP does not materially affect any the theoretical or empirical results. The reason why we use the Gaussian initialisation in the main text to refer to SP is ***purely for easier comparison*** with $\mu$PC in Table 1, since we can contrast them both on the Gaussian variance. We thank the reviewer for pointing out this imprecision and now clarify in the main text that SP can refer to any of these initialisations.
> * Regarding the ill-conditioning of the inference landscape, Figure 4 should be compared with Figure A.21, which shows the ill-conditioning of ResNets ***without $\mu$PC***. Together, these figures show that the ill-conditioning of ResNets does not significantly vary with or without $\mu$PC. Figure 2 shows results for standard MLPs with no skip connections (please note that the caption makes reference to Figure A.21). The reason for this order of presentation is to highlight the trade-off between the stability of the forward pass and the ill-conditioning discussed in Section 5. Namely, we start with MLPs which are not extremely ill-conditioned but have unstable forward passes. This then lead us to ResNets which can have stable forward passes but become much more ill-conditioned. The camera-ready version of the paper will discuss the significance of the ill-conditioning of the inference landscape and $\mu$PC in more detail, and we are happy to elaborate on this further.
>
> **Summary**
>
> To summarise, we agree with the reviewer that the main limitation of our work remains the simple datasets on which $\mu$PC was tested and that this is one of the most important next research directions. We hope that we clarified the significance of the results by making clear the results of Figures A.17-8. We appreciate the reviewer’s suggestions to improve the clarity of the paper and will aim to incorporate these in the camera-ready version.
>
> **References**
>
> Innocenti, Francesco, et al. "Only strict saddles in the energy landscape of predictive coding networks?" NeurIPS (2024).

---

> > ### Comment · Reviewer_ijZ5 · 2025-08-04
> > **Thanks**
> >
> > I have read the authors' rebuttal, which has addressed my earlier questions. Although I still have reservations regarding whether this method will lead to improved learning performance on more challenging tasks, I am now convinced that this work makes a significant contribution to the predictive coding literature. Consequently, I have increased my confidence score.

---

### Official Review · Reviewer_aWaW · 2025-07-03

**Clarity:** 4
**Significance:** 3
**Originality:** 3
**Rating:** 5
**Confidence:** 3

**Summary:**

The authors present an analysis of standard PC showing that it is susceptible to vanishing and exploding inference dynamics in networks of increasing size. They reveal a trade-off between conditioning of the inference landscape and the stability of the forward pass, and introduce muPC, showing equivalence to BP, helping to bridge the gap in performance at networks of scale.

**Questions:**

This would be a stronger accept with more diverse experiments and some reflection on limitations and possible extensions of this approach. Motivation could also be developed as well as further comment on the link between this and other local learning algorithms, the possible benefits of training via local learning , limitations of backprop etc. beyond the bioplausibility issue

**Ethical Concerns:**

["NO or VERY MINOR ethics concerns only"]

**Final Justification:**

I have increased my confidence in this assessment in light of the authors' response.

**Limitations:**

No. I imagine experimental results would lend further insight into limitations and advantages of this approach. Can we expect such networks to match bp performance exactly , given theorem 1? Are there other caveats worthy of consideration.

**Quality:**

3

**Strengths And Weaknesses:**

This was a very impressive paper that was well motivated with great theoretical contributions. This work has impact not only for general literature in bioplausible learning algorithms, but by extension other forward-only and local learning algorithms and by extension novel hardware and training paradigms. Overall very impressive work!

I see only one notable weakness, which is limited experimental results. It would be interesting to showcase the direct applicability of this approach to problems of greater complexity, although training such models is surely expensive. Authors might also consider broadening the discussion of the implications of this work for other kinds of local learning algorithms, although it is alluded in the introduction.

---

> ### Author Rebuttal · Authors · 2025-07-28
>
> We thank the reviewer for their feedback and are glad that they share our excitement about the work. Below we address the key points made by the reviewer.
>
> **Limited experiments:** First, we completely agree that the main weakness of the paper are the limited experiments testing $\mu$PC, especially in terms of range of tasks and datasets explored. The main reason for this is that, as correctly noted by the reviewer, training of such deep models is very computationally expensive on GPUs even on simple datasets like MNIST, due to both the size of the networks and the sequential nature of PC inference. This cost further increases for generation tasks where more inference steps tend to be needed. For these reasons, we think that trying to extend our results, in the first instance, to convolutional networks trained on more challenging classification datasets is an important (and exciting) next research step, as mentioned in the conclusion.
>
> **Clarification on $\mu$PC and BP:** Second, we would like to clarify a potential misunderstanding regarding the relationship between $\mu$PC and BP explored in Section 6. Theorem 1 shows that under certain conditions $\mu$PC will compute the same gradients as BP ***at the initialisation of the weights***, with no guarantees about what happens during training. Figure 6 shows that this result indeed holds at initialisation but breaks down after a few steps of training in the regime of large width and depth we are interested in. This means that $\mu$PC computes (approximately) the same gradients as BP ***only*** at initialisation (at large width relative to depth). Importantly, this does not necessarily mean that $\mu$PC will underperform BP, and in fact the empirical results show that it is equally performant. As shown by Innocenti et al. (2024), for example, PC can have different gradients than BP that still allow successful (and sometimes even faster) training. Nevertheless, it is still not entirely clear why muPC is so successful, and more theoretical work is needed as noted in the text (lines 234-5). We now make these points clear in Section 6 to avoid any potential confusion.
>
> **Motivation of local learning:** The reviewer remarks that the study of local learning algorithms as alternatives to standard BP could also be better motivated beyond their biological plausibility. We agree and now mention in the introduction that these algorithms also hold the promise of more energy efficient AI and have been argued to outperform BP in more biologically relevant settings including online and continual learning (Song et al., 2024).
>
> **Expanded limitations and implications:** We agree with the reviewer that, due to limited space, our discussion of the limitations and implications of the work (including connections with other local learning algorithms) was very brief. There are two main implications that we see for other local algorithms which the reviewer specifically asks about. First, our results clearly highlight that the stability of the forward pass is as critical for PC as for standard BP, suggesting that other local algorithms could benefit from a $\mu$P-inspired parameterisation. Second, as noted in the conclusion (lines 252-4), our analysis of the inference landscape can be applied to any other algorithm performing some kind of inference minimisation. As noted, we include a preliminary investigation of equilibrium propagation in the Appendix (A.2.5).
>
> The camera-ready version of the paper will also include an expanded Conclusion section addressing the following main points: (i) why $\mu$PC is successful despite the ill-conditioning, (ii) whether $\mu$PC seems to satisfy other Desiderata than #1, (iii) the limitations of $\mu$PC and whether it is optimal, and (iv) additional future research directions. More specifically, we argue that:
> * $\mu$PC seems to work despite the ill-conditioning of the inference landscape because its forward pass initialises the activities much closer to an inference solution than standard PC (see Fig. A.11), which we show tends to vanish/explode;
> * $\mu$PC seems to satisfy not only Desideratum 1, but also Desideratum 2 about the stability of the forward pass during training. While we have no theoretical results on the training dynamics of $\mu$PC, the fact that $\mu$PC shows good performance throughout training strongly suggest that the forward pass remains stable;
> * $\mu$PC is unlikely to be the optimal parameterisation in that it was not developed specifically for PC, and more theoretical work is required in this respect.
>
> **References**
>
> Innocenti, Francesco, et al. "Only strict saddles in the energy landscape of predictive coding networks?" NeurIPS (2024).
>
> Song, Yuhang, et al. "Inferring neural activity before plasticity as a foundation for learning beyond backpropagation." Nature Neuroscience (2024).

---

### Note · Authors · 2025-08-11

We would like to thank all the reviewers for their feedback, which we believe helped improve the quality of our paper. In these final remarks, we would like to provide a global summary of the reviews and discussions, along with some additional comments.

All the reviewers acknowledged:
1. the motivation of scaling biologically plausible, local learning algorithms and PC in particular,
2. our theoretical and empirical analysis of why deep networks are challenging to train with PC, and
3. our proposed approach and the significance of our findings.

The reviewers also generally appreciated the presentation of the paper, and most imprecisions or confusions were clarified in the rebuttal. With one single exception that we return to below, we addressed all of the reviewers' concerns and questions.

A common concern was the limited experimental evaluation of our proposed parameterisation on simple datasets. As noted in some of our rebuttals, we acknowledge that this is the main limitation of our work, and it was mainly due to the very high compute cost of training such deep networks with PC compared to backpropagation. To be more precise, due to the sequential nature of PC inference, training a 100+ layer network with PC is more than 2 orders of magnitude more expensive than backprop (given the use of as many inference steps as number of hidden layers). Nevertheless, should the paper be accepted, we will aim to add some preliminary experiments on CIFAR-10, as suggested by one reviewer.

Another reviewer raised concerns about the equivalence of our proposed parameterisation ("muPC") to Depth-muP. As noted in our comments, we agree that muPC is not *exactly* equivalent to Depth-muP in one specific scaling (of the learning rate), and we appreciate the reviewer’s perspective of wanting to remain true to the original theory given the prominent framing of the paper in terms of muP. However, it is our position that our use of the term is justified for the following three reasons:
* our parameterisation uses exactly the same weight multipliers for ResNets that are specific and unique to Depth-muP;
* the two original Depth-muP papers differ in their learning rate scaling recommendation for Adam (relevant to our case); and
* one specific study (Noci et al., 2024) performed experiments with a *different learning rate scaling than the one prescribed by their Depth-muP version* and still interpreted their results to support the theory (despite the different scaling).

---

### Decision · Program_Chairs · 2025-09-17

**Decision:**

Accept (poster)

**Comment:**

This paper introduces µPC, a parameterisation for predictive coding (PC) networks inspired by Depth-µP, enabling the training of deep residual PCNs (up to 128 layers). The authors show that standard PC struggles with vanishing/exploding inference dynamics at scale, and µPC alleviates these issues, achieving stable training, competitive performance, and zero-shot transfer of learning rates across depths and widths. The work provides both theoretical analysis and empirical results, positioning itself as a step toward scaling biologically plausible local learning algorithms to modern deep architectures.

The strengths of the paper are clear. Reviewers highlighted its strong motivation (aWaW), clear theoretical framing of why PC fails at depth (ijZ5), and novel application of µP scaling to PC, which is an important and original contribution (qKFu). The work convincingly demonstrates that stable training is possible with very deep PCNs, a milestone that had not been shown before (N2W2).

The main weakness, consistently noted by reviewers, is the limited experimental evaluation, restricted to simple benchmarks such as MNIST and Fashion-MNIST. This raises questions about generalisability to more complex datasets. Concerns were also raised about the precise equivalence to Depth-µP (N2W2). During rebuttal, the authors clarified scaling choices, added analysis, and acknowledged limitations, committing to adding CIFAR-10 experiments in the final version.

On balance, despite the limited empirical scope, the methodological advance and thorough theoretical treatment make this an impactful and timely contribution. I recommend acceptance.